# Investigating the sign of stratocumulus adjustments to aerosols in the global storm-resolving model ICON

Emilie Fons[1], Ann Kristin Naumann[2,3], David Neubauer[1], Theresa Lang[2,3], and Ulrike Lohmann[1]

[1]Institute for Atmospheric and Climate Science, ETH Zürich, Zürich, Switzerland
[2]Max Planck Institute for Meteorology, Hamburg, Germany
[3]Meteorological Institute, Center for Earth System Research and Sustainability (CEN), Universität Hamburg, Hamburg, Germany

**Correspondence:** Emilie Fons (emilie.fons@env.ethz.ch)

**Abstract.**

Aerosols can cause brightening of stratocumulus clouds, thereby cooling the climate. Observations and models disagree on the magnitude of this cooling, partly because of the aerosol-induced liquid water path (LWP) adjustment, with climate models predicting an increase in LWP and satellites observing a weak decrease in response to increasing aerosols. With higher-resolution global climate models, which allow the simulation of mesoscale circulations in which stratocumulus clouds are embedded, there is hope to start bridging this gap. In this study, we present boreal summertime simulations conducted with the ICOsahedral Non-hydrostatic (ICON) global storm-resolving model (GSRM). Compared to geostationary satellite data, ICON produces realistic cloud coverage in the stratocumulus regions, but these clouds look cumuliform and the sign of LWP adjustments disagrees with observations. We investigate this disagreement with a causal approach, which combines timeseries with knowledge of cloud processes, allowing us to diagnose the sources of observation-model discrepancies. The positive ICON LWP adjustment results from a superposition of processes, with an overestimated positive response due to (1) precipitation suppression, (2) a lack of wet scavenging, and (3) cloud deepening under a weak inversion, despite (4) small negative influences from cloud-top entrainment enhancement. We also find that precipitation suppression and entrainment enhancement occur at different intensities during the day and the night, implying that daytime satellite studies suffer from selection bias. This causal methodology can guide modelers on how to modify model parameterizations and set-ups to reconcile conflicting studies concerning the sign and magnitude of LWP adjustments across different data sources.

## 1 Introduction

Stratocumulus clouds cover 20 % of the Earth in the annual mean (Wood, 2012). Stratocumulus cloud cover is particularly large over the eastern part of subtropical oceans, where they form the so-called "semi-permanent stratocumulus decks". Because these clouds are liquid low-level clouds located over dark oceanic surfaces, they efficiently reflect incoming shortwave (SW) radiation but have small effects on outgoing longwave (LW) radiation. They are therefore crucial to cool the Earth. However, the radiative properties of stratocumulus clouds will be affected by climate change, not only due to increasing temperatures in the context of global warming, but also due to anthropogenic aerosol emission reductions in the context of air pollution mitigation

(Forster et al., 2021). The IPCC predicts that subtropical marine low-level cloud cover and reflectivity will decrease in response to future warmer temperatures, leading to a positive climate feedback of $0.2 \pm 0.16$ W m$^{-2}$ K$^{-1}$, i.e., an additional warming effect. The clouds' response to globally changing aerosol emissions is more uncertain. Aerosols can modify liquid clouds by acting as cloud condensation nuclei (CCN), thereby increasing the cloud droplet number concentration $N_d$ and reducing their effective radii $r_{eff}$, at an initially constant liquid water path (LWP). The increased total droplet surface area ($\propto N_d r_{eff}^2$) makes clouds instantaneously brighter (Twomey, 1977). The shift in cloud droplet size distribution can subsequently trigger precipitation suppression, leading to prolonged cloud lifetime and increased LWP and cloud cover ($C$) (Albrecht, 1989), and/or cloud-top entrainment enhancement (Ackerman et al., 2004; Wang et al., 2003; Bretherton et al., 2007), leading to evaporation of cloud droplets and reduced LWP and $C$.

Observations (e.g., satellite data) and global climate models (GCMs) both agree that, since pre-industrial times, aerosol emissions have generated an overall cooling effect, with a total aerosol-cloud radiative forcing of -0.84 [-1.45, -0.25] W m$^{-2}$ (Forster et al., 2021). Efforts to reduce future anthropogenic aerosol emissions will therefore generate an additional warming due to reduced cloud brightness. However, there are concerns that the models might be right for the wrong reasons, as observations and models typically disagree on the sign and/or magnitude of LWP adjustments to aerosol perturbations (Quaas et al., 2009; Wang et al.; Michibata et al., 2016; Neubauer et al., 2017; Malavelle et al., 2017; McCoy et al., 2020). Especially in stratocumulus regions, GCMs tend to predict LWP increases due to increasing aerosols (e.g., Toll et al., 2017), implying a dominance of precipitation suppression. On the contrary, satellite analyses show a negative $N_d$-LWP relationship, implying a dominance of cloud-top entrainment enhancement (e.g., Gryspeerdt et al., 2019; Possner et al., 2020). However, correlative satellite studies might overestimate negative LWP adjustments by ignoring meteorological confounding. Several studies have used causal methods (Pearl, 2009) to remove spurious biases from aerosol-cloud interactions, either by directly targeting environmental confounders (Gryspeerdt et al., 2016; Varble, 2018), by simultaneously removing multiple confounding influences through multivariate regressions (e.g, Andersen et al., 2017; Wall et al., 2022), or by using opportunistic experiments to indiscriminately remove all environmental confounding (Christensen et al., 2022). For instance, Toll et al. (2017, 2019) and Chen et al. (2022, 2024) used satellite images of ship tracks and volcanic eruptions to demonstrate that LWP adjustments of low-level marine clouds to aerosols are close to zero or slightly negative. Even when environmental confounding is removed from satellite studies of LWP adjustments, they still disagree with the positive adjustments seen in some GCM simulations.

Because of their coarse horizontal grid resolution ($\approx 100$ km), GCMs are not able to resolve the dynamics of stratocumulus clouds. The sub-grid processes that mediate aerosol effects on stratocumulus clouds have to be parameterized, leading to large intermodel variability and partly explaining why model estimates of aerosol-cloud interactions continue to disagree with observations. Thanks to progress in computing techniques, a new generation of kilometric-scale climate models, called global storm resolving models (GSRMs), has emerged (Satoh et al., 2019). Inter-comparison experiments, like the DYAMOND (DYnamics of the Atmospheric general circulation Modeled On Non-hydrostatic Domains) initiative by Stevens et al. (2019), have already been conducted to evaluate the representation of cloud processes in GSRMs. GSRMs still need to rely on parameterizations of sub-grid scale turbulence, cloud microphysics and radiation, but most of them can do without convective or cloud cover parameterizations (Hohenegger et al., 2023), making them particularly suited to study deep convection in tropical regions (e.g.,

Judt et al., 2021; Nugent et al., 2022; Lang et al., 2023). For stratocumulus clouds, the spatial scales of interest include the sub-meter scale (cloud microphysics, cloud-top turbulence), the kilometric scale (large scale eddies, vertical transport) and the mesoscale (large-scale deck morphology). Although convection in stratocumulus regions is too shallow to be fully captured at kilometric resolutions, GSRMs start to bridge the gap between the physical process scale and the modeling scale for these regions as well by resolving mesoscale circulations (Stevens et al., 2020). Given the increasing popularity of GSRMs and their future use for years-long warming scenarios (e.g., as part of the "Next Generation Earth System Modelling Systems" or nextGEMS project[1]), it is crucial to study how low-level clouds are represented in these models. Heim et al. (2021) showed that the DYAMOND models were able to produce realistic low cloud cover and characteristics of the marine boundary layer (e.g., inversion, subsidence) in the South-East Atlantic stratocumulus cloud deck. Nevertheless, the ensemble displayed large inter-model variability and systematic biases with respect to observations, in particular due to the turbulent mixing schemes. Furthermore, due to computational cost considerations, the DYAMOND models all used one-moment cloud microphysics schemes, precluding an investigation of aerosol effects on low clouds as $N_d$ is not parameterized.

In this study, we present the outputs of the global GSRM ICON (Zängl et al., 2015), run at 5km-resolution with Sapphire physics (Hohenegger et al., 2023), including a two-moment cloud microphysics scheme (Seifert and Beheng, 2006) (c.f. methods in Sec. A1-A2). We look at the stratocumulus microphysical properties and their interplay with the boundary layer and the cloud macrophysics over a 45-day period (27th June, 2021 to 10th August, 2021) in the four main stratocumulus regions (Fig. 1a-d): the North-East Pacific (NEP), the South-East Atlantic (SEA), the South-East Pacific (SEP) and the North-East Atlantic (NEA). We run two simulations, one with 'low CCN' (fixed CCN concentration of 250 cm$^{-3}$ in the lower troposphere), and one with 'high CCN' (1700 cm$^{-3}$). We also evaluate the model against geostationary satellite observations and reanalysis data (Benas et al., 2023; Walther and Straka, 2020; Huffman et al., 2023; Hersbach et al., 2018a, b) (c.f. methods in Sec. A3) in order to assess the realism of the stratocumulus cloud properties in the ICON GSRM (described in Sec. 2.1).

It is challenging to directly compare aerosol-cloud interactions in the satellite and model data as the simulations are not nudged to observations. As will be discussed in Sec. 2.2, the direct comparison of emerging statistical correlations between aerosols and clouds can be tricky to interpret as they can result from different meteorological backgrounds. Instead, in Sec. 3, we use the causal methodology described in Fons et al. (2023) to disentangle superimposed processes occurring in aerosol-perturbed clouds, like precipitation suppression or entrainment enhancement. This focus on physical processes, rather than statistical associations, allows us to compare the response mechanisms of stratocumulus clouds in the model and in the observations, while removing some confounding originating from the effect of meteorology on entrainment or precipitation. The causal methodology used here is based on a causal graph of LWP drivers, which was developed using prior domain knowledge of stratocumulus cloud processes ('expert' causal graph), without using any causal discovery algorithms (Runge et al., 2023). This causal graph can be understood as a mass balance representation of the sources (condensation) and sinks (entrainment, precipitation) of LWP and can be used to detect the responses of these sources and sinks when stratocumulus clouds are perturbed by aerosols. We combine the physical knowledge contained in this graph with satellite and GSRM timeseries. The high temporal resolution of the datasets ($\Delta t = 15$ min) allows us to use the precedence of cause with respect to effect and get

[1]https://nextgems-h2020.eu/

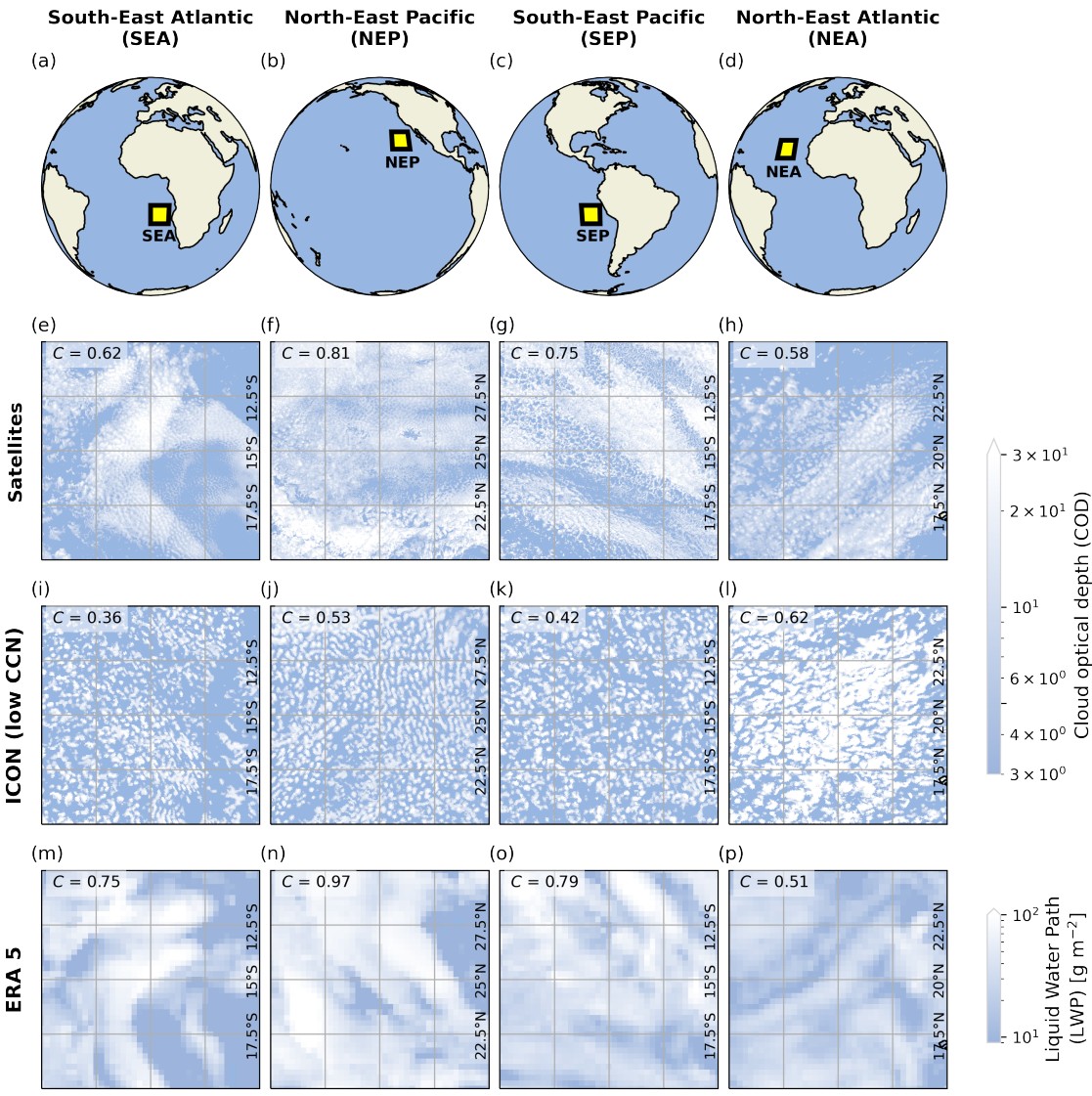

**Figure 1.** Low-level cloud cover $C$ in stratocumulus regions. **(a-d)** show the locations of the four regions, **(e-h)** show bird eye's view snapshots (06/07/2021 15:00 local time) of stratocumulus clouds as seen by geostationary satellites, **(i-l)** and **(m-p)** show snapshots at the same timestep as simulated by ICON and ERA5, respectively. Satellite and ICON cloud covers are rendered using the cloud optical depth (COD) variable, while LWP is used for ERA5 data (COD cannot be calculated with ERA5's outputs). Lower thresholds of COD=4 and LWP=9 $\mathrm{g\,m^{-2}}$ are used for the sake of this figure. Note that the ICON GSRM is not nudged to the observations, so we do not expect to see the exact same cloud formations at the same timestep. Maps from cartopy (Met Office, 2010 - 2015).

comparative causal insights into aerosol-cloud interactions. In that sense, the causal graph is used as a model evaluation tool (Nowack et al., 2020) to diagnose deviations of the model from observations.

## 2 Stratocumulus clouds in ICON-GSRM

### 2.1 How realistic are ICON's stratocumulus clouds?

Figure 1 shows snapshots of the low-level cloud cover $C$ in the four stratocumulus regions. Even though these images are snapshots, they are quite representative of the typical cloud organization over the time period of this study. ICON produces realistic $C$ in the four stratocumulus regions (Fig. 1i-l). The mean $C$ in the different regions ranges from 70 to 85 % in the

 satellite data and 55 to 80 % in the ICON data (Fig. 2a). However, even when the simulated $C$ is similar to the observed $C$ (e.g., Fig. 1h, l), the mesoscale cloud organisation looks very different in the model and in the observations. The satellite snapshots show how stratocumulus clouds are highly aggregated in closely connected cells, whether they are closed cells (Fig. 1e, f), open cells (Fig. 1g) or a mix of closed and disorganized cells (Fig. 1h) (Wood and Hartmann, 2006). ICON simulates more individual, less connected cells, i.e., clouds that seem to have cumuliform features. Increasing the CCN concentrations

 did not significantly change the overall $C$ (Fig. 2a) or cloud organization in the four stratocumulus regions (not shown). Figure 1m-p shows the ERA5 $C$ on the same day and same time step as Fig. 1e-h. Even though similar cloud structures appear in the observations and the reanalysis data, the coarser ERA5 $C$ does not agree with the observed $C$ on a pixel-by-pixel basis. This has implications for the co-location of cloud variables from the satellite and reanalysis datasets and explains why we average the data into $0.5°$x$0.5°$ grid boxes in the following analyses (see methods in Sec. A3).

 Figure 2 b-j shows mean values of cloud and boundary layer properties for the ICON simulations and the geostationary satellite data for 10 different variables: $C$, $N_\mathrm{d}$, $r_\mathrm{eff}$, cloud depth ($H$), cloud-top entrainment rate ($w_\mathrm{e}$), surface precipitation (RR), cloud-base vertical velocity ($w_\mathrm{CB}$), LWP, boundary layer height (BLH) and estimated inversion strength (EIS) (see methods for calculations). While $C$, $r_\mathrm{eff}$, $H$, $w_\mathrm{e}$, LWP and BLH are of the same order of magnitude between the model and the observations, there is a significant underestimation of $N_\mathrm{d}$ and a significant overestimation of RR by ICON compared

 to the satellite data. Due to the horizontal grid discretization, both ICON and ERA5 underestimate the variability of in-situ vertical velocities (Fig. 2g), with maximal $w_\mathrm{CB}$ much smaller than real updraft speeds at stratocumulus cloud base ($\approx 0.5$ m s$^{-1}$; Wood, 2012). While Fig. 2g shows the vertical velocity variance in the coarse $0.5°$x$0.5°$ data, Suppl. Fig. 1 shows the same plot but using the original 5-km ICON data, confirming the low bias for cloud-base updraft speeds in this ICON set up. The underestimation and low variability of $N_\mathrm{d}$ in ICON is directly explainable by the low vertical velocities, as the

 CCN activation is parameterized as a function of the (fixed) CCN concentration and the vertical velocities at cloud base (Segal and Khain, 2006). The underestimated $N_\mathrm{d}$ leads to slightly overestimated $r_\mathrm{eff}$ and strongly overestimated precipitation rates, both at cloud-base and at the surface. In the high CCN experiment, $N_\mathrm{d}$ and $r_\mathrm{eff}$ are shifted to more realistic values compared to observations, although it should be noted that the high CCN concentration amounts to 1700 cm$^{-3}$, which is unrealistically high for marine regions. Even though cloud-base precipitation decreases by a factor of $\approx 2$ in the high CCN experiment, precipitation

 still remains 1 to 2 orders of magnitude higher than the satellite-measured precipitation, indicating a systematic high bias for stratocumulus precipitation in this ICON model configuration with respect to satellite-derived precipitation estimates.

Additionally, there is a slight underestimation of EIS in ICON compared to ERA5. It can be noted that ERA5 itself might underestimate the real EIS in stratocumulus regions: Although model developments led to an increasingly better representation

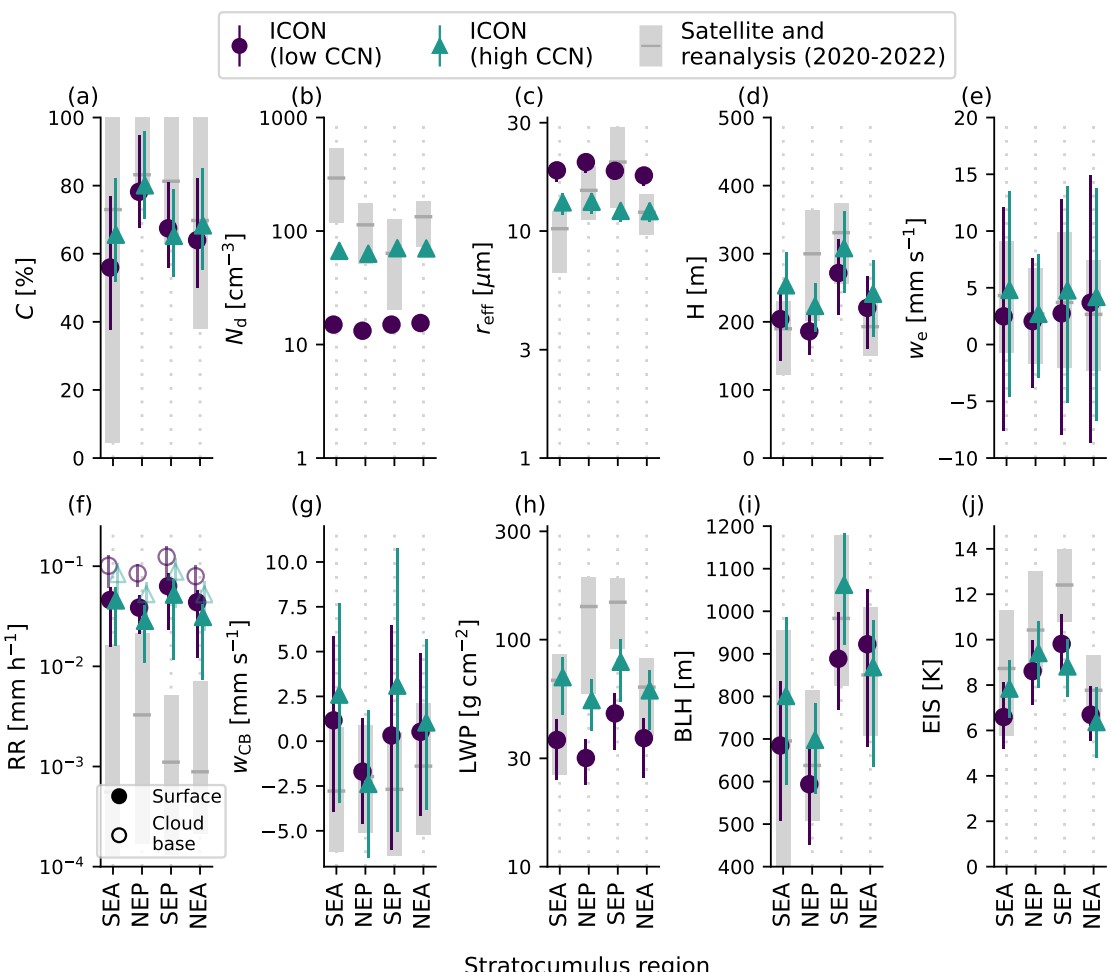

**Figure 2.** Cloud and boundary layer properties for the four stratocumulus regions. The markers show the average value for ten different variables: cloud cover **(a)**, cloud droplet number concentrations **(b)**, cloud droplet effective radii **(c)**, cloud depth **(d)**, cloud-top entrainment rate **(e)**, rain rate **(f)**, cloud-base updraft **(g)**, liquid water path **(h)**, boundary layer height **(i)**, estimated inversion strength **(j)**. The vertical bars (purple, green and thick grey) show the spread between the $25^{\text{th}}$ and the $75^{\text{th}}$ percentile of the instantaneous 0.5° grid points (for the low ICON CCN, high ICON CCN and satellite/reanalysis data, respectively). The grey bar for satellite data covers three years (from 01/07 to 09/07 for 2020, 2021, 2022 vs. 2021 only for ICON) to allow for a comparison of the model values to a climatological spread. For the grey variables, $w_{\text{e}}$, BLH and EIS are derived from ERA5, while the other variables are derived from satellite products. $N_{\text{d}}$, $r_{\text{eff}}$ $H$ and LWP correspond to daytime conditions only, all other variables include both daytime and nighttime data points.

of boundary-layer stratocumulus clouds in ERA reanalysis (Köhler et al., 2011), biases remain (Ahlgrimm et al., 2018), and
the sharpness of the inversion at the top of the boundary layer is often underestimated (Kalmus et al., 2015; Zheng and Miller, 2022).

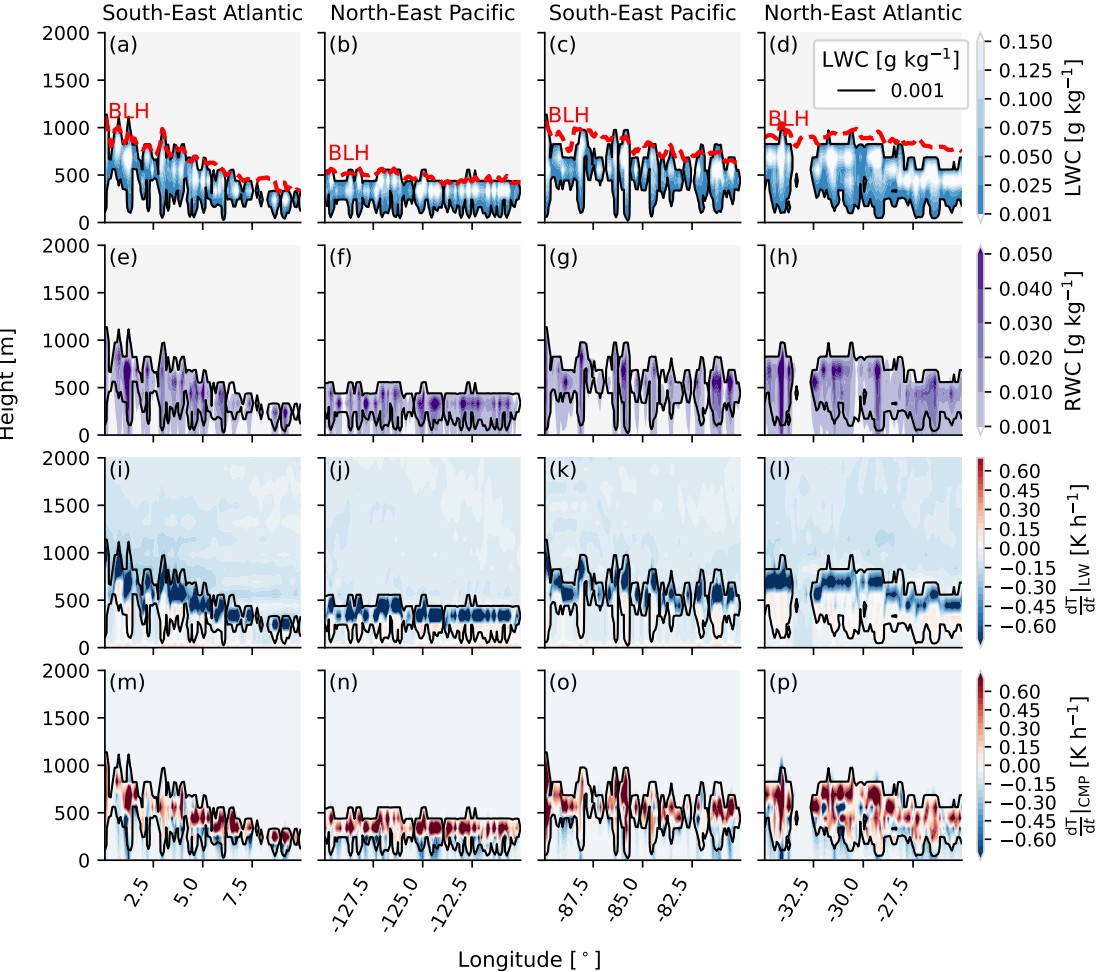

**Figure 3.** Cross-section snapshots of cloud properties simulated by ICON. The cross-sections are taken along the middle latitude of each region, and at the same timestep as Fig. 1. The first row **(a-d)** shows the cloud liquid water content (LWC) in all four regions, the second row **(e-h)** shows the rain water content (RWC), the third row **(i-l)** shows the temperature tendency due to longwave (LW) radiation and the fourth row **(m-p)** shows the temperature tendency due to phase changes caused by the cloud microphysics (CMP) scheme. On the first row, the red dotted line indicates the boundary layer height (BLH).

Figure 3 shows the vertical structure of the clouds in the low CCN simulation, at the same timestep as in Fig. 1, while Fig. 4 shows average vertical profiles for selected variables in ICON and in ERA5. ICON's clouds are well confined in the lower part of the troposphere, at the top of the boundary layer (Fig. 3 and Fig. 4e). From the smooth all-sky vertical profiles in Fig. 135    4, it could look like the boundary layer is uniformly topped by stratocumulus clouds. In reality, the boundary layer can be

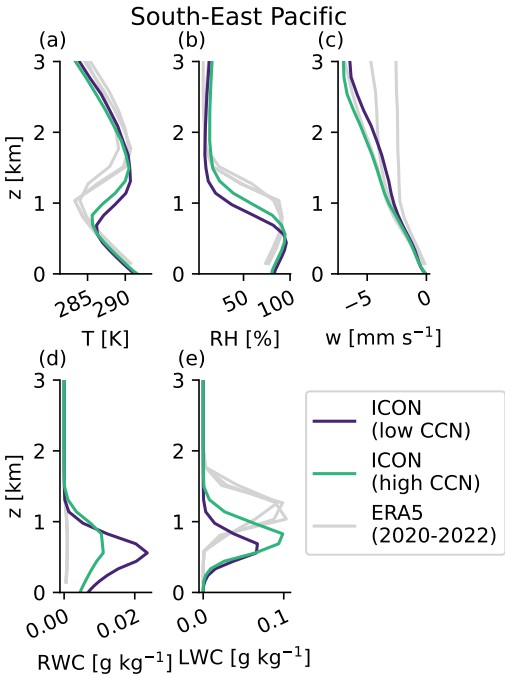

**Figure 4.** Vertical profiles of cloud and boundary layer properties. Temperature T **(a)**, relative humidity RH **(b)**, vertical velocity $w$ **(c)**, rain water content RWC **(d)** and cloud liquid water content LWC **(e)** are shown for the SEP region. These average profiles are calculated directly from the original resolution of the ICON and ERA5 data and include both daytime and nighttime all-sky conditions. Results are similar for the three other regions, and available in Suppl. Fig. 2.

decoupled from the surface and feature two cloud layers, but we find that well-mixed boundary layers represent the majority of data points in our ICON simulations (Suppl. Fig. 3). On average, the top of the boundary layer is marked by a temperature inversion of a few Kelvin (Fig. 4a), which is slightly weaker than the inversion in the reanalysis data. The free troposphere is characterized by large-scale subsidence and low relative humidity (Fig. 4b,c).

The ICON clouds seem to have cumuliform features: Even though, on average, the cloud height is similar to the observed height (Fig. 2d and Fig. 4e), the depth of individual cloud cells is highly variable, with small and deep cells that seem rather disaggregated. Figure 3a-d and Fig. 4e show that the liquid water content of the clouds increases with height in the cloud, in line with cloud adiabaticity assumptions (Lohmann et al., 2016), even though it decreases again close to cloud top, suggesting cloud-top entrainment. Figures 3e-h and 4d show the rain water content of the clouds. In the four regions, most cloud cells

precipitate, with strong precipitation coming from the deepest cloud cells. Even though some precipitation evaporates (blue color in Fig. 3m-p), there is surface precipitation across most of the domain. ICON stratocumulus clouds have LWC profiles that are similar to ERA5 profiles but RWC profiles that are several orders of magnitude larger than the reanalysis values (Fig. 4d). Although ERA5's LWC and RWC are obtained from parameterizations and are thus subject to uncertainties, this also

agrees with the lower rain rates measured by satellites (Fig. 2f), and seems to confirm a high stratiform precipitation bias in this ICON version. As expected, the clouds produce LW radiative cooling at cloud-top (negative temperature tendencies in Fig. 3i-l) due to efficient emission of LW radiation to space through the dry and cloud-free free troposphere. In stratocumulus clouds, this cloud-top cooling is an important driver of convective and turbulent mixing (Wood, 2012). Figure 3m-p shows the temperature tendency due to phase changes caused by the cloud microphysics scheme, with a positive tendency indicating net condensation, and a negative tendency indicating net evaporation. Net condensation is visible inside clouds, while net evaporation of rain water occurs below cloud base. Net evaporation of cloud water is also visible in-cloud, especially when the cloud cells are deep (e.g., Fig. 3m), indicating that evaporation of cloud/rain water also occurs inside the cloud due to entrainment.

Overall, the cloud characteristics shown in Fig. 1-4 indicate that ICON simulates rather realistic boundary layer properties and low-level cloud characteristics in the stratocumulus regions. In particular, specific features like extensive cloud cover at the top of the boundary layer and cloud-top radiative cooling are well represented by ICON. Additionally, ICON-GSRM seems to be able to generate reasonable values of cloud-top entrainment (Fig 2e), which is a key process for the water budget and evolution of stratocumulus clouds. However, there are systematic biases with respect to observations, namely too low cloud droplet number concentrations due to low vertical velocities at cloud base, and very high rain rates. These biases are bound to affect the radiative budget of the stratocumulus regions and should be carefully studied before using the GSRM in warming scenarios. In the next sections, we look at how these biases affect the LWP adjustments of stratocumulus clouds to aerosols.

## 2.2   LWP adjustment to aerosols

LWP adjustments to aerosols are typically quantified by the derivative of LWP with respect to an aerosol proxy $A$ (e.g., $A$ = Aerosol Optical Depth): $\frac{d\,LWP}{d\,A} = \frac{\partial LWP}{\partial N_d}\frac{\partial N_d}{\partial A}$. Given the difficulty of obtaining a satellite-derived aerosol proxy A that correlates well with CCN concentrations (e.g., Stier, 2016), the two partial derivatives on the right-hand side of this equation are often evaluated separately. Here, we focus on the first term, $\frac{\partial LWP}{\partial N_d}$, and how it is impacted by aerosol-driven entrainment and precipitation responses. Separating the terms and focusing only on the first one also allows for a fairer comparison of the satellite and model adjustments, as the model's $\frac{\partial N_d}{d\,A}$ term will not be defined due to the constant CCN assumption.

Figure 5 shows that LWP values are positively correlated with $N_d$ values in ICON data but negatively correlated in satellite data. This suggests that, like in GCMs, and contrary to observations, ICON simulates LWP increases in response to aerosol increases. However, correlation does not mean causation. For more insight into actual aerosol effects on the LWP, we can also look at the comparison between the low CCN and the high CCN experiments conducted with ICON. Figure 2h confirms that the LWP responds positively to increases in CCN and $N_d$ in the model, increasing confidence in the fact that this is indeed a causal response. Figure 4e shows that this increase in LWP can be divided into an increase in LWC and an increase in cloud depth. At this point, it is important to note that Fig. 4 (and 6) show all-sky vertical averages over space and time, so that differences between the low and high CCN cases could be (partly) due to modified cloud fractions/lifetimes in the high CCN case. However, Jiang et al. (2006) showed no cloud lifetime effects for cumulus clouds. Histograms of instantaneous grid point

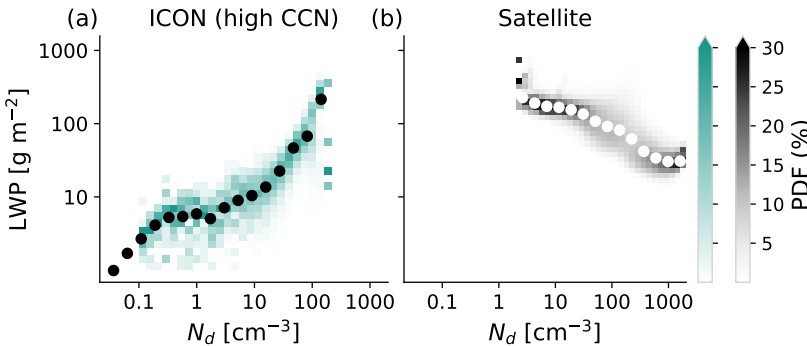

**Figure 5.** 2D histograms of LWP vs. $N_{\rm d}$. The subplots show the relationship in the model data **(a)**, and the satellite data **(b)**. The data plotted here correspond to the daytime $0.5°$ in-cloud data, including all four stratocumulus regions. See Suppl. Fig. 4 for histograms disaggregated by stratocumulus region and histograms for the 'low CCN' experiment. The color shading shows the probability density function (PDF), after binning the data points in narrow intervals of $N_{\rm d}$ values.

LWC values are also provided in Suppl. Fig. 5 and confirm that the differences between the low and high CCN simulations are still present when a potential lifetime effect is taken out.

It has been postulated that the positive stratocumulus LWP adjustments in GCMs are due to the coarse resolution and lack of explicit parameterization of size-dependent entrainment processes (Karset et al., 2020). On the contrary, the explicit parameter­ization of the autoconversion rate of cloud droplets into raindrops as a function of cloud droplet number concentrations encodes a clear causal link from $N_{\rm d}$ to precipitation in GCMs and in the ICON GSRM. As a consequence, aerosol-induced increases in $N_{\rm d}$ will cause precipitation reductions and LWP build-up in climate models. It is clear from the results presented above (Fig. 2f, 3e-h, 4d) that clouds precipitate too readily in ICON, with rain rates one or two orders of magnitude larger than the ones recorded by the satellite precipitation product or estimated by ERA5. In this highly precipitating regime, aerosol-induced shifts in the cloud droplet distribution can cause reductions in precipitation rates and result in strong absolute increases in LWP. The comparison of Fig. 4d and Fig. 6a,b shows how the precipitation profile shifts concomitantly with the droplet number and droplet size profiles when the CCN concentration is increased. Another noteworthy parameterization choice (or lack thereof) in this ICON set-up is wet scavenging. Because CCN concentrations are kept constant, the causal link for wet scavenging is turned off. Wet scavenging has been shown to induce a negative correlation between $N_{\rm d}$ and LWP (McCoy et al., 2023), so turning it off biases the correlation towards more positive values. These two mechanisms alone (precipitation suppression and no wet scavenging) could explain the positive sign of LWP adjustments in the model. In the rest of this paper, we focus on other hypotheses for this observation-model discrepancy, namely: entrainment effects and cloud deepening.

How is cloud-top entrainment simulated in the ICON GSRM? Contrary to GCMs, sub-kilometric Large-Eddy Models (LEMs) have been able to simulate negative entrainment influences on the LWP (Ackerman et al., 2004; Wang et al., 2011). With a resolution that is much better than GCMs, but still coarser than LEMs, GSRMs could be expected to simulate entrain­ment processes via the turbulence scheme, thereby driving aerosol-induced LWP reductions. The net LWP increase that is

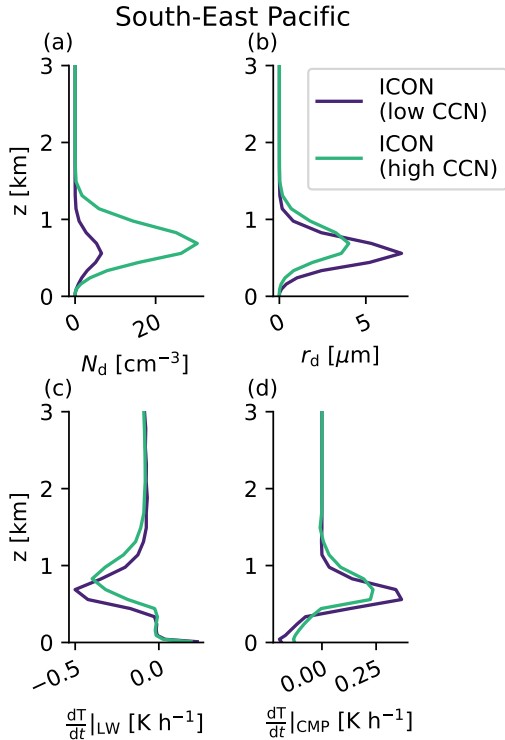

**Figure 6.** Vertical profiles of cloud microphysical and radiative properties. The subplots were obtained with the same methodology as those from Fig. 4 and show: the cloud droplet number concentration **(a)**, the cloud droplet radius $r_{\rm d}$ **(b)**, the temperature tendency due to LW radiation **(c)** and the temperature tendency due to phase changes caused by the cloud microphysics scheme **(d)**. Only the SEP region is shown here, results are similar for the 3 other regions (Suppl. Fig. 2). These profiles are all-sky averages, but instantaneous grid-point values are shown as histograms in Suppl. Fig. 5.

observed in the model does not necessarily mean that there is no entrainment response. Figure 2e does show that cloud-top entrainment rates are slightly enhanced in the high CCN simulation. Figure 6c,d gives more insight into potential mechanisms

for entrainment enhancement in ICON with the vertical profiles of the temperature tendencies related to LW radiation and phase changes in the cloud microphsyics (CMP). However, there is no cloud droplet sedimentation in our model set-up, because it is assumed to be a negligible sink for cloud water given the vertical extent of the grid boxes. Therefore, the only possible mechanisms left to explain cloud-top entrainment enhancement are: increased cloud-top LW radiative cooling and/or enhanced evaporative cooling, both resulting in increased cloud-top turbulence (Bretherton et al., 2007). Figure 6c shows that,

despite the increased LWP in the high CCN experiment, cloud-top LW radiative cooling (negative $\frac{\mathrm{dT}}{\mathrm{d}t}\big|_{\mathrm{LW}}$) is not enhanced: It is even reduced. That can be explained by the higher cloud tops in the high CCN simulation (Fig.4e), resulting in colder LW radiation emission temperatures. This eliminates the LW radiative cooling hypothesis. Figure 6d shows how net cloud water condensation leads to latent heat release (positive $\frac{\mathrm{dT}}{\mathrm{d}t}\big|_{\mathrm{CMP}}$) within the cloud. In the high CCN case, in-cloud $\frac{\mathrm{dT}}{\mathrm{d}t}\big|_{\mathrm{CMP}}$ is

reduced compared to the low CCN case. This behavior can also be seen in the instantaneous temperature tendencies in Suppl.
Fig. 5. A possible explanation is larger latent heat consumption from enhanced evaporation in the high CCN case. This means that the evaporation hypothesis is a plausible mechanism for entrainment enhancement in ICON. Interestingly, because of saturation adjustments in the model, there is no possible mechanism for droplet size-dependent evaporative rates. Therefore, the enhanced evaporation is purely due to the increase in LWP caused by precipitation suppression, i.e., there is more water left in the cloud to evaporate. This can generate more in-cloud turbulence and enhance cloud-top entrainment. This means that the real mechanisms for entrainment enhancement and the mechanisms at play in the model might be different, and this explains why we will use two different causal graphs (one for satellite data and the other for model data) in the next section.

Cloud-top entrainment is not explicitly parameterized in the ICON model used here. Therefore, it is not as straightforward to evaluate the direction entrainment responses to aerosol perturbations as it is for precipitation influences via the autoconversion or wet scavenging parameterization choices. For example, it is possible that, as expected for aerosol-perturbed stratocumulus clouds, there is entrainment enhancement and resulting cloud evaporation, but that this process is weaker than precipitation suppression, resulting in net LWP increases when CCN are increased. It is also possible that entrainment is enhanced but that it drives cloud deepening by raising the cloud top (Wood, 2007), resulting in an enhanced positive LWP response. Another difficulty is that the ICON simulations are global, and CCN concentrations were increased globally from the low to high CCN case. This means that the boundary conditions (subsidence, inversion) of the four stratocumulus regions might change between the low and high CCN experiments. For instance, the entrainment response to CCN increases might be driven by the coupling of stratocumulus cloud processes with the large-scale circulation (e.g., the Walker circulation, Dagan et al., 2023) and not directly driven by local CCN increases.

Moreover, the mechanism for cloud deepening (Fig. 2d, 4e) is not immediately obvious. Cloud deepening could be driven by large-scale meteorological changes, or it could be locally driven by entrainment under a weak inversion. Because the clouds have cumuliform features in ICON, it is also possible that other processes, such as cloud deepening from delayed precipitation (Seifert et al., 2015), or aerosol-induced warm cloud invigoration (Douglas and L'Ecuyer, 2021), are driving cloud deepening and LWP increases on top of the precipitation response.

The above considerations show how difficult it is to draw causal conclusions from simple correlations or experiment comparisons. In the next section, we apply a causal graph for LWP drivers (Fons et al., 2023) to disentangle the LWP response to increases in aerosols and cloud droplet number concentrations. This will allow us to answer the following questions concerning the model data: (1) Are cloud responses driven by local aerosol perturbations or non-local processes? (2) Does an entrainment response to aerosols exist alongside the precipitation response? (3) What is the sign of this entrainment response? (4) Do additional processes, like cloud deepening (driven by entrainment, warm cloud invigoration or rain delays), also have an importance?

 ## 3 Causal diagnosis of LWP drivers

### 3.1 Precipitation and entrainment

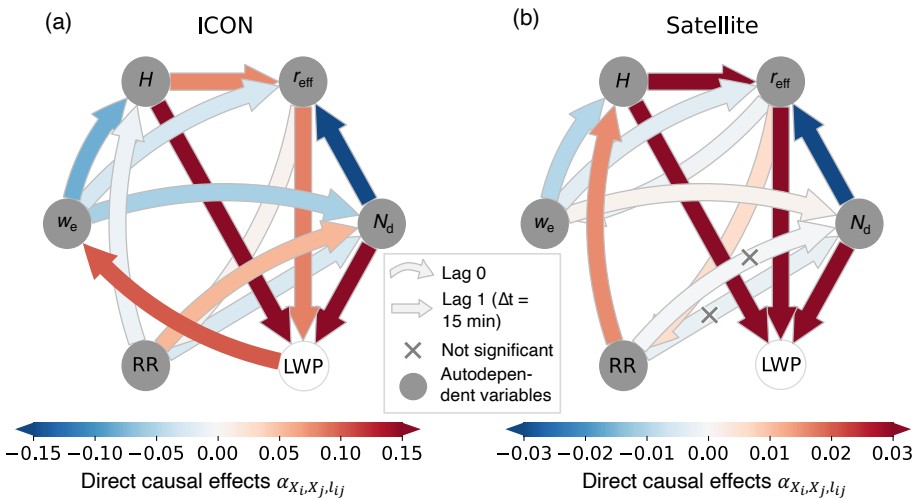

**Figure 7.** Direct causal effects for LWP drivers in stratocumulus clouds. **(a)** is the causal graph applied to the ICON data set, and **(b)** to the satellite data set. In **(a)**, one arrow has been moved compared to **(b)** due to the different mechanisms possible for entrainment enhancement in reality and in the model set-up. The straight arrows indicate contemporaneous (lag 0) effects, while the curved arrows indicate lagged (lag $1 = \Delta t = 15$ min) effects. The color shades of the arrows indicate the sign and magnitude of the causal effects, with different scales for **(a)** and **(b)**. The small grey crosses indicate the arrows that are not significantly different from 0 based on a 90 % bootstrap confidence interval.

Figure 7 shows the assumed $N_\mathrm{d}$-LWP causal graph used in the analyses. A causal graph encodes domain knowledge by qualitatively describing causal relationships (arrows) between variables of interest (nodes). Each directed arrow describes a causal effect and not merely a statistical association, and can be justified by an underlying physical process. The colors of the arrows show the magnitude and sign of the direct causal effects $\alpha_{X_i,X_j,l_{ij}}$, as calculated from the model data (Fig. 7a) and the satellite data (Fig. 7b). The graphs for the model data and for the satellite data are different (see arrow pointing to $w_\mathrm{e}$) due to the different mechanisms that are possible for entrainment enhancement in reality and in the model set-up. Both graphs describe how contemporaneous (lag 0, straight arrows) and lagged (lag $1 = \Delta t = 15$ min, curved arrows) responses in precipitation and entrainment processes can lead to LWP adjustments following aerosol perturbations. The descriptions of physical processes underlying individual arrows are given in Table 1. The causal graphs presented here contain one additional arrow (from RR to $H$) compared to the graph presented in Fons et al. (2023), due to the consideration of another physical process, cloud deepening via rain delays.

Each $\alpha_{X_i,X_j,l_{ij}}$ in Fig. 7 corresponds to the regression coefficient of $X_i$ on $X_j$ along the $l_{ij}$-lagged arrow after any confounding indicated by the causal graph has been removed. See Fig. A1 in the methods section for the example calculation of the $\alpha_{N_\mathrm{d},r_\mathrm{eff},0}$ direct causal effect. A negative (positive) direct causal effect means that an increase in $X_i$ leads to a decrease

**Table 1.** Association of the direct causal effects $\alpha_{X_i,X_j,l_{ij}}$ with the corresponding cloud physical processes. Almost all arrows are correctly detected, i.e., their sign agrees well with the direction of the underlying physical process.

| Arrow (Fig. 7) | Physical description | Expected sign | Correctly detected in satellite? | Correctly detected in model? |
|---|---|---|---|---|
| $\alpha_{N_d,\mathrm{LWP},0}$, $\alpha_{r_{\mathrm{eff}},\mathrm{LWP},0}$ and $\alpha_{H,\mathrm{LWP},0}$ | Definition of LWP | + | Yes | Yes |
| $\alpha_{N_d,r_{\mathrm{eff}},0}$ | Due to mass balance for a given LWP | - | Yes | Yes |
| $\alpha_{H,r_{\mathrm{eff}},0}$ | Condensational growth with height | + | Yes | Yes |
| $\alpha_{r_{\mathrm{eff}},w_e,1}$ | Entrainment suppression by droplet sedimentation | - | Yes | N/A |
| $\alpha_{\mathrm{LWP},w_e,1}$ | Entrainment enhancement by evaporation of cloud droplets | + | N/A | Yes |
| $\alpha_{w_e,N_d,1}$, $\alpha_{w_e,r_{\mathrm{eff}},1}$ and $\alpha_{w_e,H,1}$ | Evaporation due to cloud-top entrainment | - | Yes (homogeneous)[a] | Yes (mix)[a] |
| $\alpha_{r_{\mathrm{eff}},\mathrm{RR},1}$ | Rain enhancement | + | Yes | Yes |
| $\alpha_{\mathrm{RR},N_d,0}$ | Cloud water removal | - | Not significant | Yes |
| $\alpha_{\mathrm{RR},N_d,1}$ | Wet scavenging/Dynamical adjustments | 0/- | Not significant | No |
| $\alpha_{\mathrm{RR},H,1}$ | Cloud deepening due to rain delay | - | Inconclusive[b] | Yes |

[a]: 'Homogeneous' and 'mix' refer to the entrainment mixing regimes described by Hill et al. (2009). [b]: Results are very sensitive to the proxies used for $H$ and RR. See text for more details.

(increase) in $X_j$ after a lag $l_{ij}$. Here, the negative $\alpha_{N_d,r_{\mathrm{eff}},0}$ describes how, when a cloud is perturbed by aerosols and $N_d$ increases, the droplet size $r_{\mathrm{eff}}$ decreases to satisfy the water mass balance at an initially constant LWP. The direct causal effects can be multiplied according to Wright's path rule (Wright, 1921) to compute total causal effects $\beta_{X_i,X_j,l}$ between two variables that are not directly linked by an arrow in the graph. For example, the effect of $N_d$ on RR (mediated by $r_{\mathrm{eff}}$) is computed with: $\beta_{N_d,\mathrm{RR},1} = \alpha_{N_d,r_{\mathrm{eff}},0} \times \alpha_{r_{\mathrm{eff}},\mathrm{RR},1} < 0$. The negative sign of $\beta_{N_d,\mathrm{RR},1}$ agrees well with the precipitation suppression mechanism that we expect to see in aerosol-perturbed clouds (Albrecht, 1989).

Almost all direct causal effects $\alpha_{X_i,X_j,l_{ij}}$ have the same sign in the satellite and the model data, although magnitudes are different (Fig. 7, Table 1). We suspect that the magnitudes in the model graph might be higher because the model data set is more self-contained than the co-located satellite/reanalysis dataset, where the variables are derived from different sources. Another explanation could be that the ICON graph captures a non-linear lower-$N_d$/higher-RR regime, potentially causing the alphas to vary in magnitude due to the linear assumption behind Wright's path analysis. For these reasons, we mostly comment on the sign of the physical processes. The signs of the satellite causal effects calculated here also agree well with the calculations from Fons et al. (2023), which were conducted on 2-year, coarser-resolution timeseries of the SEA region only.

Aerosol-induced increases in $N_d$ cause entrainment enhancement both in the satellite data and in the model. In the satellite graph, this is described mathematically by $\beta_{N_d,w_e,1} = \alpha_{N_d,r_{\mathrm{eff}},0} \times \alpha_{r_{\mathrm{eff}},w_e,1} > 0$. This describes how, following aerosol in-

creases, reductions in $r_{\mathrm{eff}}$ will slow down the sedimentation of cloud droplets at the cloud top, where more efficient longwave and evaporative cooling of the smaller droplets will generate turbulence and enhance entrainment of free tropospheric air into the boundary layer. In the model data, entrainment enhancement is directly described by $\alpha_{\mathrm{LWP},w_e,1} > 0$. This is different from the satellite graph, as explained in 2.2, and is due to the lack of cloud droplet sedimentation and to the saturation adjustment step in the CMP scheme. This means that there is no size-dependence of cloud droplet sedimentation, LW radiative cooling and evaporation in the model. Instead, the arrow from LWP to $w_e$ indicates that, in the model, enhanced entrainment is purely due to the increase in LWP (caused by precipitation suppression), i.e., there is more water left in the cloud to evaporate from a water budget perspective. This will cause evaporative cooling, generate more in-cloud turbulence and enhance cloud-top entrainment.

Entrainment of warm and dry free-tropospheric air into the cloud has similar effects on cloud droplets in the satellite data and in the model (see arrows from $w_e$ to $N_d$ and $r_{\mathrm{eff}}$): evaporation. In the model, both $N_d$ and $r_{\mathrm{eff}}$ are reduced, indicating a mix of homogeneous and inhomogeneous entrainment regimes (Hill et al., 2009). The satellite data display a more homogeneous regime with a reduction in $r_{\mathrm{eff}}$ but not $N_d$, indicating that the mixing timescale is longer than the evaporation timescale, allowing the entrained air to mix with the cloud and evaporate all cloud droplets homogeneously. The effect of $w_e$ on $N_d$ is even slightly positive. On the contrary Fons et al. (2023) found a mix of homogeneous and inhomogeneous entrainment regimes. The difference might arise due to the different regions and seasons considered in the studies. In particular, the months of July and August correspond to the start of the biomass burning season on the western coast of North America and in southern Africa. Smoke aerosols frequently overlie stratocumulus clouds and can be mixed into the cloud layer via entrainment (e.g., Redemann et al., 2021), potentially explaining a positive effect of $w_e$ on $N_d$, while the effect on droplet sizes and cloud depths is still negative due to evaporation. After the entrained air has led to the evaporation of cloud droplets, the latent heat consumption by evaporation generates cooling and turbulence, leading to further entrainment enhancement. Mathematically, this feedback loop can be confirmed in the satellite graph by following the path from $w_e$, through $H$ and $r_{\mathrm{eff}}$ and then back to $w_e$: $\beta_{w_e,w_e,2} - \alpha_{w_e,w_e,1} \times \alpha_{w_e,w_e,1} = \alpha_{w_e,r_{\mathrm{eff}},1} \times \alpha_{r_{\mathrm{eff}},w_e,1} + \alpha_{w_e,H,1} \times \alpha_{H,r_{\mathrm{eff}},0} \times \alpha_{r_{\mathrm{eff}},w_e,1} > 0$ ($\alpha_{w_e,w_e,1} \times \alpha_{w_e,w_e,1}$ is the causal autodependency component of the total causal effect). It can be noted that the formula is more complicated for the model graph because of the mediation by RR which delays the effect.

The causal quantification of precipicitation influences is more elusive. In Fons et al. (2023), the lag 0 arrow from RR to $N_d$ was found to be negative, indicating cloud water removal by precipitation, while the lag 1 arrow was close to zero. We hypothesized that the lag 1 arrow was the superposition of a negative process (CCN scavenging by precipitation; Grandey et al., 2014) and a positive process (speculatively linked to dynamical adjustments of updraft speeds below cloud base when cold pools are present; Terai and Wood, 2013). In this study, both the lag-0 and lag-1 arrows are found to be negative but insignificant, i.e., the confidence interval includes 0. This might indicate that the noise-to-signal ratio is not favorable with this satellite precipitation proxy. In the model data, the lag-0 arrow from RR to $N_d$ is accurately detected to be negative (cloud water removal), but the lag-1 arrow is detected to be positive. This is because there is no wet scavenging in the model, as CCN concentrations are constant in space and time: Instead, other dynamical adjustments might cause the lag-1 arrow to become positive. Interestingly, when we replace cloud-base precipitation with surface precipitation rates in the ICON data, the lag-0

arrow becomes positive and the lag-1 arrow becomes negative (Suppl. Fig. 6a). This seems to indicate that the quantification of precipitation processes is very sensitive to the chosen precipitation product and to the choice of causal time lags. Because the satellite proxy quantifies surface precipitation, and surface precipitation from stratocumulus is hard to retrieve from space (e.g., Zhu et al., 2022), precipitation processes might be inaccurately estimated. Additionally, although 15 min is a good

temporal resolution to capture stratocumulus cloud updrafts (see methods), it might be too coarse to resolve faster precipitation influences, potentially resulting in spurious positive precipitation influences in the model data (Runge, 2018). It should also be noted that the causal method used here is based on a linearity assumption which might be broken for non-linear precipitation processes. In particular, the lagged arrow from RR to $N_d$ (wet scavenging and dynamical effects of precipitation) might be incorrectly captured in case of non-linear threshold effects of precipitation on the dynamics of the boundary layer. The onset of

precipitation (arrow from $r_{eff}$ to RR) is also non-linear, but is expected to follow a power law, thus it should be well captured thanks to the log-transformation of the variables (see methods). Finally, hidden confounding variables (i.e., not yet included in the causal graph), like relative humidity (Grandey et al., 2014), might also further bias the arrows from RR to $N_d$ and will need to be evaluated in future research. For a complete description of the other arrows in the causal graph, see Fons et al. (2023).

The application of the causal graph shows that low-level clouds in ICON respond as expected to aerosol perturbations, with

325 precipitation suppression and cloud-top entrainment enhancement. This answers questions (1), (2), and (3) as enumerated at the end of the previous section: (1) we do detect local aerosol effects on the ICON clouds; (2) entrainment is enhanced even without cloud droplet sedimentation; (3) entrainment enhancement leads to LWP reductions due to evaporation, even though these reductions are masked in the net positive LWP adjustment (Fig. 2h).

## 3.2 Cloud deepening

The causal processes described above do not explain why cloud depth seems to increase with aerosols in ICON. In fact, we could even predict cloud thinning due to entrainment-related evaporation, as indicated by the blue arrow from $w_e$ to $H$ in both the satellite and the model data. Cloud thinning by entrainment can happen under certain thermodynamic conditions (Wood, 2007) and has been simulated in limited area models (Ackerman et al., 2004; Bretherton et al., 2007). However, cloud thinning by entrainment can be compensated by other processes, potentially causing a net deepening effect. This invites us to think of

new mechanisms to explain cloud deepening: warm cloud invigoration (Douglas and L'Ecuyer, 2021) or aerosol-induced rain delays and subsequent cloud layer growth (Seifert et al., 2015; Vogel et al., 2016).

Warm cloud invigoration occurs because condensational growth on numerous and smaller droplets is more efficient than condensational growth on larger droplets, thereby generating additional latent heat and turbulence, which can drive increases in updraft speeds. This can cause cloud deepening and increase $N_d$ due to the dependence of CCN activation on supersaturation

(Segal and Khain, 2006). However, such a mechanism is not possible in this model set-up due to saturation adjustments in the CMP scheme. In case of supersaturation, the saturation adjustment scheme condenses all the excess water vapor, independently of the cloud droplet number concentration or radius. Instead, we look at how aerosol-induced rain delays might affect the cloud depth: as aerosols reduce droplet sizes and precipitation is suppressed, clouds deepen to produce rain and maintain a so-called subsiding radiative-convective equilibrium (SCRE) that is imposed by the balance of the large-scale subsidence and evaporative

and convective tendencies (Stevens and Seifert, 2008; Seifert et al., 2015; Rosenfeld et al., 2019). LEM experiments by Vogel et al. (2016) show that, as precipitation is reduced (or even suppressed), convective mixing moistens the inversion if the subsidence is weak, resulting in boundary layer growth (seen in three regions in Fig. 2i) and cloud deepening. Considering the low subsidence in most of the study regions (Fig. 4c and Suppl. Fig. 2) and the weak inversion strength (Fig. 2j), the rain-delay hypothesis is a likely process to explain the cloud deepening as seen in ICON.

The arrow from RR to $H$ allows us to test the rain-delay hypothesis (Fig. 7). In the model data (Fig. 7a), the lag-1 arrow from RR to $H$ is found to be negative, describing how early precipitation onset prevents further cloud deepening. In other terms, when precipitation is suppressed due to aerosol increases, smaller droplets can be lifted higher as they grow in the updraft region, resulting in cloud deepening under a weak inversion. One immediate consequence is LWP build-up, as $\alpha_{N_{\mathrm{d}},r_{\mathrm{eff}},0} \times \alpha_{r_{\mathrm{eff}},\mathrm{RR},1} \times \alpha_{\mathrm{RR},H,1} \times \alpha_{H,\mathrm{LWP},0} > 0$. Eventually, the cloud deepening can allow the cloud to form precipitation ($\alpha_{H,r_{\mathrm{eff}},0} \times \alpha_{r_{\mathrm{eff}},\mathrm{RR},1} > 0$)

and satisfy the SCRE imposed by large-scale forcing.

    The arrow from RR to $H$ is found to be strongly positive in the satellite data (Fig. 7b). A positive effect of precipitation on cloud depth seems physically unlikely (Wood, 2007), therefore we hypothesize that this is due to the violation of the adiabaticity assumption used to compute $H$ from satellite retrievals when clouds are precipitating. The same arrow is found to be weakly negative when $H$ is estimated from ERA5, in agreement with the ICON data, but weakly positive when both $H$ and

RR are estimated using ERA5 (Suppl. Fig. 6b). Christensen and Stephens (2011) used satellite data to evaluate aerosol effects on stratocumulus cloud deepening, and found no significant cloud deepening in closed cell regimes, while it could occur for open cell regimes in unstable boundary layers. Given the prevalence of closed cell morphologies in the stratocumulus regions considered here (e.g., Wood and Hartmann, 2006, or Suppl. Fig. 7), one could have expected a null effect of aerosols on cloud depth in the satellite data. However, given the dependence on the choice of cloud depth and precipitation proxies, the effect of

RR on $H$ in the satellite data is marked as 'Inconclusive' in Table 1.

    The analysis of the RR to $H$ arrow answers question (4), i.e.: are additional processes, like cloud deepening, also important for LWP adjustments in ICON? We confirm that aerosol increases can result in cloud deepening due to rain delays under a weak inversion in the model, contributing to positive LWP adjustments. In the satellite data, it is not as clear whether cloud deepening contributes to LWP build-up due to the uncertainty surrounding both precipitation retrievals and cloud depth

estimations. Analysis of other data sources, e.g., in situ field data might help to answer this question.

### 3.3   Temporal developments

Figure 8 shows how the precipitation and entrainment responses develop with time. Temporal developments $\beta_{X_i,Y_i,l}$ are not calculated from the observed temporal evolution of cloud fields over several hours, but instead through the propagation of direct causal effects and autodependency coefficients, using Wright's path tracing formula (methods). Importantly, even if

24-hour cloud developments are considered, they are still calculated from the multiplication of coefficients computed over 15 min increments over which we assume negligible advection of cloud fields past the 0.5° grid boxes. This propagation is done assuming stationarity of the causal effects throughout the day and night. In particular, even when direct causal effects are

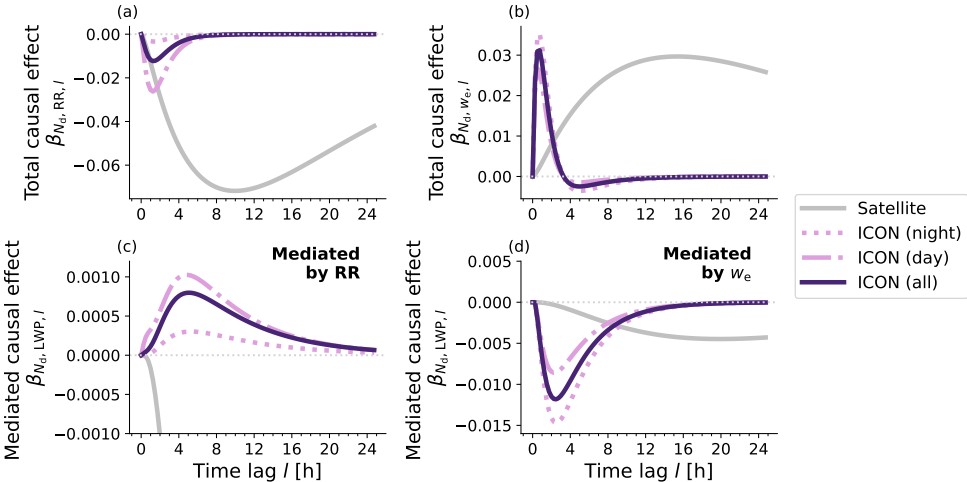

**Figure 8.** Temporal developments of causal effects. The subplots show the temporal developments of **(a)** the total causal effect of $N_\mathrm{d}$ on RR, **(b)** the total causal effect of $N_\mathrm{d}$ on $w_\mathrm{e}$, **(c)** the RR-mediated causal effect of $N_\mathrm{d}$ on LWP, and **(d)** the $w_\mathrm{e}$-mediated causal effect of $N_\mathrm{d}$ on LWP, after an initial positive perturbation in $N_\mathrm{d}$ at $l = 0$. In each subplot, the grey and purple lines correspond to the two causal graphs presented in Fig. 7a and b, respectively. Note that the vertical axis of subplot **(c)** is cropped at -0.001 and therefore cuts the grey line. This is to ensure that the other lines are not squished.

calculated from daytime satellite data only, propagation throughout the night is still possible by assuming that nighttime and daytime direct causal effects are equal (see methods and Fons et al., 2023, for more details)

The temporal developments are consistent with the direct effects discussed above: $\beta_{N_\mathrm{d},\mathrm{RR},l}$ is consistently negative, in both the satellite data and the model data, indicating precipitation suppression (Fig. 8a). Interestingly, $\beta_{N_\mathrm{d},RR}$ is weaker in the model than in the satellite data. This too-weak dependence of RR on $N_\mathrm{d}$ could explain why precipitation rates do not decrease much when increasing CCN to $1700\,\mathrm{cm}^{-3}$ in the high CCN experiment (Fig. 2f).

  In the satellite data, $\beta_{N_\mathrm{d},w_\mathrm{e},l}$ is consistently positive, indicating long-lasting cloud-top entrainment enhancement (Fig. 8b).

In the model data, the initial positive $\beta_{N_\mathrm{d},w_\mathrm{e},l}$ becomes slightly negative after three hours, indicating that cloud-top entrainment first increases, then decreases slightly instead of continuing to increase. This different model behavior is probably explained by the limited mechanisms for entrainment enhancement that are possible in the model.

  Figure 8c,d shows the fraction of the total effect of $N_\mathrm{d}$ on LWP that is mediated by RR and $w_\mathrm{e}$, respectively. Note that these effects are not direct and immediate but mediated and delayed. The mediated effect of RR is computed along the causal

graph path that starts from $N_\mathrm{d}$, goes through $r_\mathrm{eff}$ and RR (precipitation suppression), and then back to $N_\mathrm{d}$ and $H$ (cloud water removal and cloud deepening) before integrating to LWP. For the satellite data, the mediated effect of $w_\mathrm{e}$ is calculated along the path that starts at $N_\mathrm{d}$, goes through $r_\mathrm{eff}$ then $w_\mathrm{e}$ (entrainment enhancement), and then back to $N_\mathrm{d}$, $r_\mathrm{eff}$ and $H$ (evaporation) before integrating to LWP (for the model, the path is longer as it goes through RR, then LWP, and then finally $w_\mathrm{e}$).

Figure 8c shows that the effect of aerosol-induced precipitation suppression on the LWP is positive (albeit weak) in the model data (purple line), in line with the expected LWP build-up from precipitation suppression. For the satellite data, the RR-mediated LWP response (grey line) becomes very negative very quickly (cropped in Fig. 8c) due to the spurious positive arrow from RR to $H$ in Fig. 7b. Negative precipitation-mediated influences on the LWP, as seen in the satellite data, are deemed to be non-physical and probably stemming from the violation of the adiabaticity assumption when clouds are precipitating and/or limitations of remote-sensing precipitation products for stratocumulus clouds (e.g., Zhu et al., 2022). The positive precipitation-mediated response of stratocumulus clouds to aerosols should mostly result from reduced cloud water removal, and to some extent from cloud deepening if the inversion is weak. However, the causal graph actually detects that the positive response mostly results from cloud deepening in the model (compare Fig. 8c to Suppl. Fig. 8). This indicates that the causal method applied here might be limited to accurately detect all precipitation processes, as already discussed in Sec 3.1.

Figure 8d shows that the effect of aerosol-induced entrainment enhancement on the LWP is negative, denoting continued evaporation due to mixing of the warm and dry free tropospheric air into the cloud. By comparing the scales of Fig. 8c-d, it would seem that the entrainment-mediated reductions in LWP should prevail compared to the precipitation-mediated build-up in the model. However, this is not possible since the net aerosol effect on LWP is positive in the model (Fig. 2h): we hypothesize that the issues associated with RR in the causal graph, as identified above, might be responsible for this too-weak mediated effect of RR.

The timescales of precipitation suppression and entrainment enhancement are much shorter for the ICON clouds, with model peak responses occurring within a couple of hours and satellite peak responses occurring after 12 hours. It would make sense that these processes occur faster in the model than in reality, as ICON simulates clouds with cumuliform features rather than stratiform features and cumulus clouds have notably shorter lifetimes (< 4-5 h, Seifert et al., 2015) than stratocumulus clouds (< 24 h, Christensen et al., 2020). However, it is possible that the longer timescales in the satellite data are due to the mismatch of temporal resolution between the co-located satellite and reanalysis products, introducing an artificial delay in the computed causal effects.

Figure 8 also shows the temporal developments computed from model daytime and nighttime data separately. The satellite data only include daytime conditions because the retrievals of cloud optical depth and $r_{\text{eff}}$ are made using visible and near-infrared wavelength channels. One assumption that we made when deriving the 24-h temporal developments from the satellite data is stationarity of causal effects as a function of the diurnal cycle. This assumption turns out to be false as aerosol-induced precipitation suppression and cloud-top entrainment enhancement occur at different intensities in the nighttime and daytime in ICON. Even though nighttime stratocumulus do not instantaneously matter for the SW radiation budget, nocturnal clouds can persist into the following day, especially in the early morning (Lu and Seinfeld, 2005). This implies that LWP adjustments inferred from daytime satellite studies, or LEM studies focusing on nocturnal clouds (e.g., Ackerman et al., 2004; Glassmeier et al., 2019) will tend to suffer from selection bias. In our ICON simulations, we see stronger aerosol-induced entrainment enhancement and weaker precipitation suppression at night. While it could make sense that entrainment enhancement could be reduced during the day due to dampened cloud-top radiative cooling by shortwave radiation absorption (Ackerman et al., 2004; Zhang et al., 2024), LEM studies have rather observed weaker effects of precipitation suppression during the day due to

low daytime absolute precipitation rates (Lu and Seinfeld, 2005; Sandu et al., 2008). The different behavior observed with our

ICON simulation could be explained by our coarser resolution and too high precipitation rates (including during the daytime). Therefore, we refrain from inferring the sign of the day vs. night selection bias for stratocumulus LWP adjustments from this simulation, but we note that it exists.

The temporal developments confirm the results of the analyses of the direct causal effects: aerosols cause precipitation suppression and associated LWP build-ups, as well as entrainment enhancement and associated LWP reductions in both the

observations and in the model. In particular, cloud deepening enhances the positive influence of RR on LWP in the model. The temporal developments also provide additional information concerning timescales, and seem to confirm that ICON clouds are shorter-lived than real stratocumulus clouds, with faster precipitation and entrainment responses to aerosols than observations. Importantly, both nighttime and daytime conditions need to be taken into account to accurately estimate LWP adjustments and their shortwave radiative effects.

## 4   Conclusions

The ICON model is able to simulate realistic low-level clouds in stratocumulus regions, even though there are systematic biases concerning cloud droplet number concentrations, precipitation rates and cloud morphologies with respect to satellite observations. We also observe that, similarly to typical GCMs, ICON simulates positive stratocumulus LWP adjustments to aerosol increases. This suggests that positive effects from aerosol-induced precipitation suppression prevail compared to

negative influences of cloud-top entrainment enhancement. There were several a priori hypotheses that could explain the observation-model differences: too strong rain rates in the model compared to observations; no wet scavenging in the model; no cloud droplet size-dependent entrainment parameterization in the model, potentially preventing negative entrainment influences on LWP; cloud deepening under a weak inversion. While the first two hypotheses can be easily studied because of the explicit parameterization (or lack thereof) of the corresponding physical processes in ICON, we used a causal graph as a diagnosis

tool to evaluate the two other hypotheses. We found that the positive response of LWP to aerosols in the model results from a superposition of different processes.

First, biases and approximations related to precipitation processes lead to a positively enhanced $N_\mathrm{d}$-LWP relationship in the model. This is due to precipitation being too strong and the absence of wet scavenging. As detected by the causal graph and as expected from Albrecht (1989), increases in $N_\mathrm{d}$ lead to precipitation suppression and LWP build-up in the model. The resulting

LWP build-up is detected to be too weak, potentially due to non-linearities of precipitation processes or a temporal resolution that is coarser than the process timescale of precipitation processes (Runge, 2018). However, it is clear that, with average rain rates one to two orders of magnitudes higher than observed rain rates, aerosol effects will cause precipitation suppression and LWP build-up due to the explicit parameterization of rain autoconversion as a function of cloud droplet number concentration in the cloud microphysics parameterization scheme (Seifert and Beheng, 2006). The absence of wet scavenging in this ICON

version is noticeable through the change of sign of the lagged arrow from RR to $N_\mathrm{d}$ in the causal graph between the two graphs

in Fig. 7. As shown by McCoy et al. (2020), wet scavenging induces negative $N_\mathrm{d}$-LWP relationships, and could therefore well explain why the $N_\mathrm{d}$-LWP relationship is positive in the model but negative in observations.

While the dependence of the sign of the $N_\mathrm{d}$-LWP relationship on precipitation suppression and wet scavenging in models is well established in the literature, entrainment influences in coarse models are more obscure. With the causal approach, we observe that cloud-top entrainment enhancement does occur in the ICON GSRM, even without a parameterization of entrainment as a function of droplet sizes. Entrainment enhancement was not a priori expected as the ICON set-up used here does not parameterize cloud droplet sedimentation, which has been identified as a key process to initiate the entrainment enhancement feedback loop (Ackerman et al., 2004; Bretherton et al., 2007). In the model, entrainment enhancement is purely driven from increased evaporation of accumulated LWP due to precipitation suppression. This is different from droplet-size dependent entrainment mechanisms that occur in observations (radiative and evaporative feedbacks) due to parameterization choices that limit the realm of what is possible in the model. Like in observations, entrainment enhancement causes evaporation of the cloud droplets, with a negative effect on the LWP.

Finally, we detect that delays in the onset of precipitation can cause cloud deepening in ICON (Seifert et al., 2015), potentially due to the model's vertical resolution, resulting in difficulties to simulate a strong inversion. In the satellite data, such a response is not as clearly detected, potentially due to the difficulty in reliably estimating cloud depths from satellite or reanalysis data. However, in a stable stratocumulus-topped boundary layer, we might not expect any vertical growth if the inversion strength is high enough, as observed by Christensen and Stephens (2011).

In summary, we used the causal approach as a diagnosis tool to decompose the positive $N_\mathrm{d}$-LWP relationship seen in the model into its physical components. We found that the relationships can be decomposed in: positive effects from precipitation suppression in clouds that precipitate too strongly; positive effects from the absence of wet scavenging; positive effects from cloud deepening; negative effects from entrainment enhancement. However, in ICON, this negative effect from entrainment is weaker than the positive effects due to the superposition of precipitation effects, the absence of wet scavenging and cloud deepening. In the near future, the coupling of the newer version of the ICON GSRM with the HAM aerosol module (Stier et al., 2005) should be finalized. For the moment, only a GCM version of ICON-HAM exists (Salzmann et al., 2022). Once the ICON-HAM GSRM is available, it would be interesting to rerun these analyses and evaluate how the results change when aerosol effects, including wet scavenging, are explicitly parameterized.

The causal analyses provided interesting results concerning the temporal evolution of aerosol effects. We observe that cloud processes occur faster in the model compared to observations, which is consistent with how the clouds look: In ICON, clouds are more disagreggated and look more cumuliform, while in reality, stratocumulus clouds are stratiform and more aggregated with a longer lifetime than cumulus clouds (Jiang et al., 2006; Christensen et al., 2020). In fact, Manshausen et al. (2022) used ship tracks to show that LWP adjustments are weakly negative in stratocumulus cloud regimes, but positive in the trade cumulus cloud regime where most tracks are 'invisible'. This is consistent with the cumuliform-looking ICON clouds showing positive LWP adjustments in our study. By comparing daytime and nighttime causal effects in ICON, we also demonstrate that aerosol-induced precipitation suppression and entrainment enhancement occur at different intensities during the day and the night. This is due to the strong diurnal cycle of stratocumulus cloud processes (Wood, 2012) and implies that satellite studies

using daytime cloud measurements (like this study), or modeling studies focusing on nocturnal stratocumulus clouds (e.g., Glassmeier et al., 2019) will suffer from selection bias (Pearl, 2009).

Of course, the conclusions of this study are dependent upon the validity of the causal graph and the reliability of the given data sets. In Fons et al. (2023), we tested several other causal graphs to evaluate the robustness of the results, allowing us to select the causal graph with the highest physical plausibility. This physical graph is adjusted for the present study to account for model-reality differences, and to add a physical process (cloud deepening due to rain delays). Despite differences in the data sets (regions, seasons, resolution), the signs of the causal effects are consistent between the previous study and this study, increasing our confidence in the methodology. However, the causal methodology seems unable to reliably estimate absolute magnitudes of causal effects, making it difficult to assess whether the entrainment response or the precipitation response dominates (compare Fig. 8c and d). This could be related to additional sources of confounding that we have not considered in this study, such as large-scale transport of air masses (Mauger and Norris, 2007) or relative humidity (Grandey et al., 2014). As demonstrated by Arola et al. (2022), correlated noises for variables retrieved from the same satellite instrument can also induce confounding in the $N_d$-LWP relationship. Such correlated noises could be an issue for the satellite graph, as $N_d$, $r_{eff}$, $H$ and LWP are derived from the same radiance measurements using the adiabaticity assumption. In addition, RR retrievals are made by combining microwave measurements from polar-orbiting satellites to infrared measurements made by the geostationary imagers that are also used for the cloud property retrievals, potentially also introducing correlated noises. Biased rain effects could also originate from the linearity assumption used in the present causal approach, and calls for an investigation of non-linear methods in future research (Runge, 2018). Non-linear causal methods could be a good option to better estimate precipitation effects but also understand how non-linear decoupling of the boundary layer can modulate the results presented here.

Even though the absolute magnitudes of the linear causal effects are somewhat unreliable, signs of causal effects can be used to diagnose the existence (and direction) of physical processes in the model, hinting at model modifications that might yield a better model-observation agreement. For example, a strong inversion is a key feature of stratocumulus regions (Wood, 2012) and might be achieved in the model by increasing the vertical resolution (Bogenschutz et al., 2023), perturbing the turbulence scheme (e.g. Possner et al., 2014) or implementing more sophisticated turbulence closure schemes (Shi et al., 2018). Such improvements could prevent cloud deepening and a shift from a stratiform to a cumuliform regime. This cloud regime shift could also reduce the high precipitation bias, and a stronger humidity gradient at the inversion could enhance the negative effects of entrainment on LWP. As mentioned above, using ICON-HAM would also be interesting to quantify the effects of wet scavenging. In addition, other modifications could be implemented, e.g.: activating cloud droplet sedimentation, as previous studies (Ackerman et al., 2004; Bretherton et al., 2007) have highlighted the key role of this process in cloud-top entrainment enhancement and resulting negative LWP adjustments; or reducing the efficiency of rain autoconversion. Implementing such changes in the model could bring model estimates of aerosol-cloud radiative forcing closer to observations and eventually increase our confidence in climate projections made with GSRMs.

*Code and data availability.* Code for the data processing and analysis is provided on Zenodo (10.5281/zenodo.10580241).

The ICON model code is available online (https://code.mpimet.mpg.de/projects/iconpublic/wiki/Instructions_to_obtain_the_ICON_model_code_with_a_personal_non-commercial_research_license). The simulation runscripts are available with the rest of the code on zenodo.

The satellite timeseries used for the analyses were generated from co-located SEVIRI (Copyright (c) (2020) EUMETSAT), GOES-R (NOAA), GPM and ERA5 data (generated using Copernicus Climate Change Service information [2022]). MODIS Level 3 data (used for comparison purposes) were downloaded from https://ladsweb.modaps.eosdis.nasa.gov/archive/allData/61/MOD08_D3. MIDAS morphology data (used for a supplementary figure) were provided by I.L. McCoy in a personal communication. GOES data were downloaded from NOAA's AWS API. SEVIRI data are freely available from https://wui.cmsaf.eu/safira, GPM data from https://disc.gsfc.nasa.gov/datasets and ERA5 data from https://cds.climate.copernicus.eu. The processed timeseries and analyses outputs are provided on Zenodo (10.5281/zenodo.10580438).

## Appendix A: Methods

### A1   ICON set-up

We use the ICOsahedral Nonhydrostatic model (ICON, Zängl et al., 2015) as a GSRM in the Sapphire configuration (Hohenegger et al., 2023), with prescribed sea surface temperatures and initial conditions taken from the European Centre of Medium Range Weather Forecast (ECMWF) analysis. ICON is run with the same set up as in Lang et al. (2023), i.e., in a global set-up, for 45 days, from the $27^{th}$ of June, 2021 to the $9^{th}$ of August, 2021. The first four days are considered model spin-up and left out of the analysis. The model time step is 40 s, and the outputs are instantaneous values with an output frequency of 15 min to enable the comparison with the satellite data. The model is run at approximately 5 km horizontal resolution. While this can seem too coarse to accurately capture stratocumulus cloud processes, Heim et al. (2021) noted that there was no significant improvement in low cloud representation when increasing the horizontal resolution from 4 km to 500 m. The vertical grid is made of 110 hybrid sigma levels between the surface and a height of 75 km. The vertical resolution increases progressively from 20 m at the surface to 400 m at an altitude of 8 km, with vertical resolutions around 125 - 160 m at the inversion level in the stratocumulus regions.

In order to evaluate aerosol effects on the stratocumulus cloud decks, we used the two-moment cloud microphysics scheme by Seifert and Beheng (2006). We ran two simulations: one with moderate CCN concentrations (250 cm$^{-3}$) and one with high CCN concentrations (1700 cm$^{-3}$), all other set up parameters being equal otherwise. Note that the CCN concentration is not uniform over the whole atmospheric column, but instead is set to the fixed value in the lower troposphere and decays exponentially with altitude. For the sake of simplicity, these 2 simulations are referred to as 'ICON (low CCN)' and 'ICON (high CCN)' in this article, even though 'low CCN' can be misleading since 250 cm$^{-3}$ corresponds to typical CCN concentrations in stratocumulus-topped marine boundary layers (Roberts et al., 2010; Allen et al., 2011; Wang et al., 2022; Howes et al., 2023). The first simulation was run for 45 days (minus four spin-up days), while the second simulation was only run for 12 days (minus four spin-up days) due to the high computing time requirements of global storm-resolving runs with a two-moment cloud microphysics scheme. The common analysis period for the 2 simulations (01.07.21 to 09.07.21) is 8-day long, while the common analysis period for the satellite data vs. the low CCN experiment is 41-day long. The shorter common period is

implicitly used throughout the article whenever the high CCN experiment is analyzed (Fig. 2, 4, 5), while the longer period is used when only the low CCN experiment is used (all of Sec. 3).

Other parameterization choices include: the 3D turbulence mixing scheme from Smagorinsky (1963) with the modification by Lilly (1962) (as implemented by Dipankar et al., 2015; Lee et al., 2022), the RTE-RRTMGP scheme by Pincus et al. (2019) for radiative transfer, and the JSBACH land model from Raddatz et al. (2007). The shallow and deep convection schemes are switched off, and we use an all-or-nothing cloud scheme, i.e., the cloud fraction is set to 0 or 1 depending on a threshold for cloud water and cloud ice content.

## A2   ICON time series

The standard outputs of the model were processed to make time series for the 6 variables of interest in this study: low-level cloud droplet number concentration $N_{\mathrm{d}}$, cloud droplet radius (called $r_{\mathrm{eff}}$ for consistency with the satellite denomination), cloud depth $H$, cloud-top entrainment rate $w_{\mathrm{e}}$, rain rate RR and liquid water path LWP. Stratocumulus clouds were identified as the uppermost cloud layer within the boundary layer. $H$ was computed as the sum of the cloudy model level heights. $N_{\mathrm{d}}$ is averaged over the cloud height, while $r_{\mathrm{eff}}$ is computed as the maximum over the cloud height. We made this choice to be consistent with the adiabatic assumption used for satellite retrievals of cloud microphysical properties. In adiabatic clouds, the $N_{\mathrm{d}}$ profile is constant with height, while the $r_{\mathrm{eff}}$ profile increases with height above cloud base, meaning that satellites will observe cloud-top droplet radii that are close to the maximum (Lohmann et al., 2016). LWP is obtained by summing the product of the LWC times the layer depth for each cloud level. LWP is defined as the cloud liquid water path only, and does not include rain water. To identify the uppermost cloud level in the boundary layer, we had to diagnose the boundary layer depth BLH. BLH was estimated as the height at which the relative humidity drops below 50 % (as in Bretherton et al., 2013). As shown in Fig. 3, this corresponds well to the cloud top. $w_{\mathrm{e}}$ was diagnosed using a boundary layer mass balance equation from Stull (1988):

$$\frac{\mathrm{d\,BLH}}{\mathrm{d}t} = \frac{\partial\,\mathrm{BLH}}{\partial t} + \boldsymbol{v} \cdot \nabla \mathrm{BLH} = w_{\mathrm{e}} + w_{\mathrm{subs}},$$

where $\boldsymbol{v}$ is the horizontal wind vector and $w_{\mathrm{subs}}$ is the large-scale subsidence rate, which we estimate as the vertical velocity at the inversion level. Like in Heim et al. (2021), we make the approximation that the instantaneous term of the derivative is negligible compared to the advective term, and we thus calculate $w_{\mathrm{e}}$ as $\boldsymbol{v} \cdot \nabla \mathrm{BLH} - w_{\mathrm{subs}}$. EIS was diagnosed following Wood and Bretherton (2006). For RR, only surface precipitation is a standard output of the model. Therefore, we diagnosed cloud-base RR a posteriori, as a function of the rain water content and the rain drop number concentrations following the parametric equations from Seifert and Beheng (2006).

## A3   Satellite time series

The simulation outputs are compared to geostationary satellite cloud retrievals obtained from the Spinning Enhanced Visible and InfraRed Imager (SEVIRI) aboard the 11[th] Meteosat satellite, and the Advanced Baseline Imager ABI aboard the 16[th] and 17[th] Geostationary Operational Environmental Satellites (GOES 16 and 17). With a central longitude of 0°, Meteosat

11 can see the South-East and North-East Atlantic, including the Namibian and Canarian Sc decks, as defined by Klein and

595 Hartmann (1993). GOES 17 sees the Californian deck in the North East Pacific (central longitude of -137.2°), while GOES 16 sees the Peruvian deck in the South East Pacific (central longitude of -75.2°). Note that the footprint resolution of these satellite products in the regions of interest is approximately 3-5 km, and depends on the latitude and longitude of the measurement point with respect to the satellite. We match the time period of the satellite data to the time period of the GSRM runs (1st of July-9th August, 2021). For SEVIRI, we downloaded the CPP and CTX products of the CLoud property dAtAset using SEVIRI

(CLAAS) 3.0 (Benas et al., 2023). For GOES, we downloaded the cloud fraction, cloud optical depth, cloud droplet effective radius at cloud top, cloud-top temperature and pressure (ABI-L2-ACMF, ABI-L2-CODF, ABI-L2-CPSF, ABI-L2-ACHTF and ABI-L2-CTPF) (Walther and Straka, 2020). Note that, for both satellites, the optical cloud properties are only available for daytime conditions. Low clouds were filtered by selecting only liquid clouds in SEVIRI and by selecting cloud top pressures larger than 680 hPa in GOES. We calculated the cloud droplet number concentration, the cloud depth and the liquid water

path using the cloud adiabaticity assumption, following Brenguier et al. (2000) and Quaas et al. (2006). For further details concerning these calculations, see Fons et al. (2023). Both GOES-ABI and Meteosat-SEVIRI data have been validated (Walther and Straka, 2020; Benas et al., 2023) against more commonly used polar-orbiting satellite instruments, like the Moderate Resolution Imaging Spectroradiometer (MODIS) aboard the Aqua and Terra satellites (Platnick et al., 2015). However, few studies specifically validate $N_d$ derivations from geostationary satellites against those from polar-orbiting satellites, so we

include in the supplementary material a comparison of GOES/SEVIRI $N_d$ to MODIS $N_d$ (Suppl. Fig. 9). The geostationary $N_d$ agrees pretty well with the MODIS $N_d$, except for a constant positive bias in the SEA region (unproblematic given the data standardization), and a non-constant positive bias in the NEA region.

We colocated the satellite data with precipitation data from the Global Precipitation Measurement (GPM) Integrated Multi-satellitE Retrievals Version 7 (iMERG V07) (Huffman et al., 2023). We then added reanalysis data from ERA5 (Hersbach

et al., 2018a, b): BLH, EIS, $w_e$, $H$. EIS was diagnosed from pressure level ERA5 data following Wood and Bretherton (2006). $w_e$ was calculated following the equation from Stull (1988) as described above. Although we already had an estimate for $H$ from the satellite data, we also diagnosed it from ERA5 as a comparison, by summing cloudy level heights in the boundary layer.

All data were colocated to the 0.25° grid of ERA5 and linearly interpolated to the temporal resolution of SEVIRI (15 min).

Because of the mismatch between the low cloud cover between ERA5 and the satellites (e.g., Fig. 1 e, m), we average the co-located data to a coarser 0.5° resolution. We assume that, at these coarser resolutions, the cloud top entrainment derived from ERA5 will be approximately co-located with the right cloud structures in the satellite data, even if the cloud structures were not strictly matching on a pixel-by-pixel basis in the high resolution data. To allow for a fair comparison between the satellite and the model data, the model time series (obtained as described above) were also averaged to a coarser 0.5° grid.

Importantly, the average for cloud properties is performed on cloudy pixels only. The graph analysis therefore only captures adjustments of in-cloud properties to aerosols, excluding CF adjustments from the analysis. For consistency with other studies in the literature (e.g., Bellouin et al., 2020), the cloud properties are log-transformed in both the satellite and model timeseries.

Note that Fons et al. (2023) carried out the same type of causal analyses as the ones carried out here, but using coarser satellite data ($10°$).

## A4 Causal inference

The causal method used here consists in applying a causal graph of LWP adjustments to time series data of cloud properties. The graph was drawn by the authors, i.e., it was derived from domain knowledge and not obtained by causal discovery (Runge et al., 2023). The time series have a temporal resolution of $\Delta t$ = 15 min, which is close to the average process timescale of air parcel movements from base to top in a stratocumulus cloud. This is ideal as it allows to resolve feedback loops and witness how changes in cloud properties (e.g., $N_{\mathrm{d}}$) propagate in time, using the precedence of cause with respect to effect (Fons et al., 2023).

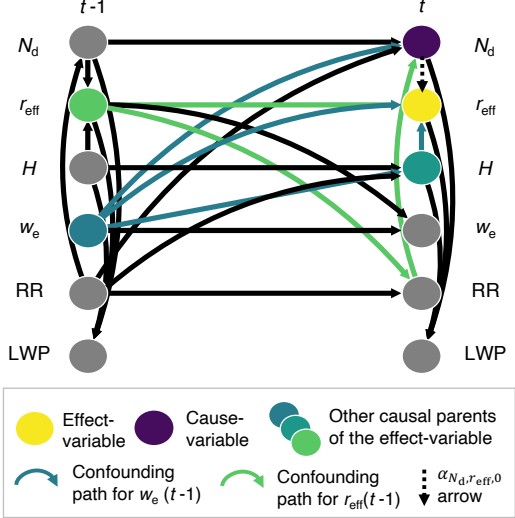

**Figure A1.** Calculations of direct causal effects using Wright's approach. This figure demonstrates how to calculate $\alpha_{N_{\mathrm{d}},r_{\mathrm{eff}},0}$ in the framework of Wright's linear approach, by carrying out a multiple linear regression of $r_{\mathrm{eff}}(t)$ on all its causal parents, including $N_{\mathrm{d}}(t)$. The other causal parents are: $r_{\mathrm{eff}}(t-1)$ and $w_{\mathrm{e}}(t-1)$, which are both confounders (see teal green and lime green arrows), as well as $H(t)$. Note that $H(t)$ itself is not a confounder but because it is a parent of the effect-variable, controlling for its effect by including it in the regression helps to reduce the variance of the results (Runge, 2021).

Following Wright's approach (Wright, 1921; Runge et al., 2015), the causal graph is used in combination with the data to detect and remove confounding influences and compute causal effects of aerosols on the LWP, as opposed to traditional linear regression coefficients. Direct causal effects $\alpha_{X_i,X_j,l_{ij}}$ refer to the causal effects between two variables that are directly connected by a directed arrow in the causal graph. Direct causal effects are calculated as partial linear regression coefficients of the effect-variable $X_j$ on the cause-variable $X_i$ in the multiple linear regression of $X_j$ on all its causal parents, i.e., those variables with arrows pointing directly towards $X_j$ in the causal graph (see an example in Fig. A1). In essence, this is similar

to the cloud controlling factor (CCF) approach (e.g., Wall et al., 2022), where causal parents are CCFs, except that the causal graph explicitly formalizes why CCFs are included (or not) in the regression by defining the expected relationships between the variables of interest. Total causal effects $\beta X, Y, l$ designate causal effects between two variables $X$ and $Y$ that are not directly connected by an arrow in the causal graph. Total causal effects are calculated from the direct causal effects using Wright's path tracing approach (sum of product rule), i.e., by tracing all the directed paths that join the two variables in the graph, and multiplying the coefficients along each path. This is similar to multiplying partial sensitivities in the CCF approach. Mediated causal effects correspond to the fraction of a total causal effect that is mediated by a given variable, by applying the path tracing formula only to those paths that flow through the mediator. In particular, the temporal developments shown in Sec. 3.3 are also calculated using the path tracing formula. For further explanation of causal effect estimation approaches, please see the methods section in Fons et al. (2023), which uses the same exact methodology. All computations and graph plots are made using the Tigramite package in Python (https://github.com/jakobrunge/tigramite).

In this study, the LWP causal graph is applied to the 41 days of the satellite and ICON (low CCN) timeseries. While in Sec. 2, we compared the low vs. high CCN simulations to infer aerosol effects, the causal methodology used in Sec. 3 infers causal effects from time variations in variables of interest, i.e, a methodologically different approach from the comparison of the low vs. high CCN simulations.

Note that all variables $X$ in the time series were corrected for the diurnal cycle prior to the causal effect computations, as the diurnal cycle can constitute a source of confounding.

$$X_{\text{corrected}}(t) = \frac{X(t) - \overline{X(t)}}{\sigma_{X(t)}},$$

where $\overline{X(t)}$ and $\sigma_{X(t)}$ are the diurnal instantaneous average and standard deviations, i.e., the average/standard deviation of X at a given time of day (e.g., 10:15 am), computed over the whole time series. The time series were not corrected for the seasonal cycle given their short duration. The causal effect computations are run for the four regions all together (i.e., aggregated), assuming the low-level clouds in these four regions obey to the same physical processes.

Confidence intervals were computed using a bootstrapping method with $n = 100$ members. Direct causal effects are considered significantly positive or negative when the bootstrap confidence interval does not include 0.

*Author contributions.* E.F. developed the concept of the study together with U.L. and D.N. T.L. and E.F. ran the ICON model simulations under the guidance of A.K.N. E.F. wrote the code for the data processing, for the causal workflow and for the data post-processing. E.F., A.K.N., D.N. and U.L. worked on to the interpretations of the results. E.F. drafted the manuscript with contributions from all other co-authors.

*Competing interests.* The authors declare no competing interests.

*Acknowledgements.* We would like to thank the anonymous reviewers, as well as the editor, Guy Dagan, for comments that improved the manuscript. We also thank Raphaela Vogel, Juan Pedro Mellado and Junhong Lee for helpful discussions on the topic of parameterizations in ICON. This work was supported by the European Union's Horizon 2020 research and innovation program under Marie Sklodowska-Curie grant agreement No. 860100 (iMIRACLI). A.K.N. and T.L. were funded by the Deutsche Forschungsgemeinschaft (DFG, German Research Foundation) under Germany's Excellence Strategy—EXC 2037 'CLICCS—Climate, Climatic Change, and Society'—Project Number 390683824.

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
