# Peer review of "Investigating the sign of stratocumulus adjustments to aerosols in the global storm-resolving model ICON"

_EGUsphere, 2024_

## Author Comment (AC1)

**Authors' answers to Anonymous Referee #1**

We thank the Anonymous Referee #1 for their valuable feedback on our manuscript, which improved the quality of the manuscript significantly. We answer their comments below, using blue font. Where necessary, we indicate the line numbers (Ln) of edits made to the manuscript like so: (Ln in original manuscript/Ln in updated manuscript).

Also please note that we found a tiny mistake in the standardization of the data and re-ran the analyses but the great majority of results remain unchanged: only one arrow in the graph goes from insignificant to significant, but it does not change sign. We added a sentence describing this arrow in the text.

This is an interesting and well-presented analysis of aerosol-cloud adjustments in a convection-permitting model over Sc decks. The analysis technique (causal graphs) is novel and is used to compare to observations. The paper is well-written. I have a few concerns that are detailed below related to the appropriateness of comparing a convection-permitting model with fixed cloud condensation nuclei to observations using this particular framework. It is my opinion that these concerns can be handled by adding caveats.

If the authors find it appropriate and not excessively time consuming, they may find that the robustness of their analysis and their suggestion that there is a strong benefit to high resolution would be aided by including analysis of a lower resolution version of the same model with interactive aerosol and with fixed CCN so that they provide examples to argue (i) that their assumption of fixed CCN is not affecting their results and (ii) that the high resolution model actually improves model performance. In particular, constant CCN makes it a bit hard to interpret the causal graphs used in this analysis. To use such a high resolution model some parts of the simulation need to be sacrificed, but the ability of the aerosol, cloud, and precipitation to couple is key to reproducing observed present-day covariability and fairly compare observations and models (Stevens and Feingold 2009; McCoy et al. 2020; Wood et al. 2012; Gryspeerdt et al. 2019). Is there a way to modify the causal graph if one arrow linking precipitation, CCN, and cloud is disabled in the model, but exists in the observations?

We agree that it would be very interesting to conduct the analyses suggested here. However, the coupling of the ICON model with the HAM aerosol module is not finalized and extensive scientific validation still needs to be carried out. This is the topic of ongoing work at ETH Zürich and is the reason why we resorted to using an ICON version with constant CCN concentrations. Results from very few ICON-HAM GCM simulations (Salzman et al., 2021; https://doi. org/10.1029/2021MS002699) have been published in the literature, but these were based on an old version of the ICON model, and it would not make sense to use this model version today. It would be interesting to conduct a full sensitivity study with respect to resolution and to the consideration (or not) of the HAM module when ICON-HAM is ready, but this is beyond the scope of this paper. We now mention these considerations in the conclusion paragraph for clarity (Ln 402 of original manuscript/Ln 478-481 of revised manuscript).

Concerning how this affects our analyses: the only causal graph arrow that would be affected by the constant CCN assumption is the lag-1 arrow from RR to $N_d$, which is expected to describe both wet scavenging and (thermo-)dynamical boundary layer adjustments to precipitation. With a constant CCN assumption, this arrow would be left to describe only the second process. This is the reason why this lagged arrow is (weakly) negative in the observations but positive in the model. Of course, the absence of wet scavenging could explain why LWP adjustments are different in the model and in the observations. The point of applying the causal graph to the constant CCN model is to evaluate other hypotheses that might also explain this difference, namely: wrongly simulated entrainment enhancement and cloud deepening.

Throughout the paper, we streamlined the discussion a bit more to make it clearer that wet scavenging is another hypothesis that can explain the different observation-model LWP adjustments. For example, we added this to the abstract: *"We find that the positive LWP adjustment to increasing aerosols in ICON results from a superposition of processes, with an overestimated positive response due to (1) precipitation suppression, (2) a lack of wet scavenging, and (3) cloud deepening under a weak inversion, despite (4) small negative influences from cloud-top entrainment enhancement"*; and we added this to section 2.2 (Ln 174/Ln 195-199): *"Because CCN concentrations are kept constant, the causal link for wet scavenging is turned off. Wet scavenging has been shown to induce a negative correlation between $N_d$ and LWP (McCoy et al., 2023), so turning it off biases the correlation towards more positive valuesThese two mechanisms alone (precipitation suppression and no wet scavenging) could explain the positive sign of LWP adjustments in the model. In the rest of this paper, we focus on other hypotheses for this observation-model discrepancy, namely: entrainment effects and cloud deepening."*, and we also mentioned the lack of an explicit parameterization for aerosol-induced entrainment enhancement and cloud deepening in our ICON version, justifying the utility of the causal graph for the study of these two mechanisms.

Ln5: Observing = inferring if it is based on correlation. If there is a transient change in aerosol it may be possible to say there is an observe a causal response. On ln 39 the cited studies are correlative, rather than causal.

Thanks to the reviewers' comments, we have realized that the discussion of causal inference methods was a bit terse. We added a few sentences to the introduction to distinguish the studies that are correlative vs. the studies that implement causal inference methods, and what some of those methods are (Ln 40/Ln 40-50):

*"[…] On the contrary, satellite analyses show a negative Nd-LWP relationship, implying a dominance of cloud-top entrainment enhancement (e.g., Gryspeerdt et al., 2019; Possner et al., 2020). However, correlative satellite studies might overestimate this negative LWP adjustment by ignoring confounding meteorological influences. Several studies have used causal methods (Pearl, 2009) to remove spurious biases from aerosol-cloud interactions, either by directly targeting environmental confounders (Gryspeerdt et al., 2016; Varble, 2018), simultaneously removing multiple confounding influences through multivariate regressions (e.g, Andersen et al., 2017; Wall et al., 2022), or by using opportunistic experiments to indiscriminately remove all environmental confounding (Christensen et al.,*

*2022). For instance, Toll et al. (2017, 2019) and Chen et al. (2022, 2024) used ship tracks and/or volcanic eruptions to demonstrate that LWP adjustments of low-level marine clouds to aerosols are close to zero or slightly negative. Even when environmental confounding is removed from satellite studies of LWP adjustments, they still disagree with the strong positive adjustments seen in some GCM simulations. [...]"*

Ln 5: While high resolution models are probably generally good since fewer processes need to be resolved, we cannot provide a climate model that resolves the micro-scale processes that lead to precipitation suppression and entrainment thinning.

We changed the abstract sentence to reflect this comment: *"With higher-resolution global climate models, which allow the simulation of mesoscale circulations in which stratocumulus clouds are embedded, there is hope to start bridging this gap."* We do not add more detail in the abstract due to the word limit, but we do mention later in the introduction that cloud microphysics, turbulence and radiation still need to be parameterized.

Ln 35: Also (Wall et al. 2022).

Thank you for the suggestion. We added this citation to the list of studies that use causal methods for LWP adjustments estimations (see two comments above), as Wall et al. (2022) control for meteorology with the cloud-controlling factor approach.

Ln 40: While I generally agree that models probably overpredict adjustments due to minimal representation of size-dependent entrainment (Karset et al. 2020). Please note that the cited papers here are correlative rather causal. As discussed later in the intro (Christensen et al. 2022) there are opportunities for dealing with causal ambiguity in observations of aerosol-cloud interactions. Also note that in (McCoy et al. 2020) both models and observations imply a negative adjustment from aerosol-cloud interactions in the present day based on similar techniques to (Gryspeerdt et al. 2019) even though the model response to anthropogenic aerosols is an increase in LWP.

As mentioned above, we modified the introduction to clearly separate correlative studies from studies that implement causal inference principles.

Following your above comment on wet scavenging, we also added a discussion of the implications of the constant CCN assumption, and how this could explain the positive $N_d$-LWP relationship, and we now explicitly cite McCoy et al. (2020) and their discussion of the negative impacts of wet scavenging in section 2.2.

We have also added the citation to Karset et al., 2020 in section 2.2 (Ln 166/ Ln 186) when discussing size-dependent entrainment parameterizations, thank you for the suggestion.

Ln 44-59: I agree with all of this, but I think it would be good to qualitatively state what the range of length scales of interest are in Sc clouds. They are still definitely sub-km (Wood 2012).

We have added a sentence describing the spatial scales of interest for the study of stratocumulus clouds, from the sub-meter scale to the mesoscale (Ln 50/ Ln 60-62).

Ln 65: While fixed CCN is useful especially in high resolution models where running full aerosol chemistry would be expensive, it is somewhat unphysical since scavenging can't remove any CCN. This is an aspect of present day covariability between clouds, precipitation, and aerosol that seems like it would be needed when comparing between observations and the model output. Discussions of the source of covariability that goes cloud->precipitation->aerosol see discussion in (Gryspeerdt et al. 2019; McCoy et al. 2020; Wood et al. 2012). It would be interesting to contrast the analysis in this study with a lower resolution version of the same model with interactive aerosol and without interactive aerosol.

Unfortunately, as mentioned above, the ICON GSRM is not available yet with interactive aerosols. We do believe LWP adjustments in ICON need to be re-evaluated when ICON-HAM is ready, to compare how the adjustments operate with and witout HAM, and at different resolutions. This has now been added as an outlook to the discussion section.

However, the appeal of the causal graph is that one can separate physical processes. While it is evident that the causal arrow from precipitation to cloud droplet numbers will be affected by the constant CCN concentration, the other arrows should not, allowing us to separately investigate other processes that might differ for LWP adjustments in the observations vs. in the model, namely: entrainment and cloud deepening. As mentioned above, we streamlined the discussion in section 2.2. to make this reasoning clearer in the manuscript.

Ln70: One confounder is that aerosol sources tend to be on land, so there is spurious correlation between lower LWP and aerosol (Wood et al. 2012; Gryspeerdt et al. 2019). I am not sure nudging really impacts that all that much and since you have set CCN to a constant through the atmosphere this source of covariance is removed. Another strong source is that thicker clouds tend to rain more- reducing Nd and creating covariance that goes from clouds to precipitation to aerosol (McCoy et al. 2020; Wood et al. 2012; Gryspeerdt et al. 2019). This is also artificially turned off if CCN is set to a constant.

Ln 90: as the authors note- the simulations are not nudged, so it is hard to know how to compare these simulation by eye in the absence of comparisons of the ICON meteorology to observations/reanalysis.

We answer these two comments together:

We have reformulated the discussion on Ln70. First, as mentioned above, we moved the discussion of confounding that was previously on Ln70 up to Ln 40. Then, we made it clear that the point of the causal graph is to focus on physical processes rather than emerging statistical relationships, i.e. allowing a comparison in terms of process sign and strength between the observations and the models (Ln 70/Ln 81-87).

*"It is challenging to do a direct comparison of aerosol-cloud interactions in the satellite data and in the model data as the simulations are not nudged to observations. As a consequence, as discussed in Sec. 2.2, the direct comparison of emerging statistical correlations between*

*aerosols and clouds can be tricky to interpret as they can result from different meteorological backgrounds. Instead, in Sec. 3, we use the causal methodology described in Fons et al. (2023) to disentangle superimposed processes occurring in aerosol-perturbed clouds, like precipitation suppression or entrainment enhancement. This focus on physical processes, rather than statistical associations, allows us to compare the response mechanisms of stratocumulus clouds in the model and in the observations, while removing some confounding originating from the effect of meteorology on entrainment or precipitation."*

To be transparent, we also added in the conclusion some sources of meteorological confounding (relative-humidity, large-scale transport) that the present study neglects for the sake of simplicity (Ln 415/Ln 500). As you mention, the fact that aerosol sources tend to be on land is an example of neglected confounding. Concerning the fact that thicker clouds tend to rain more and scavenge more aerosols, this should be captured in the causal graph through the path $H \rightarrow r_{\text{eff}} \rightarrow RR \rightarrow N_d$, with the exception that, in ICON, the RR $\rightarrow N_d$ arrow only describes dynamical adjustments since wet scavenging is turned off. This is described in section 3.1. of the manuscript.

Ln 100: Agree- ERA5 is just running reanalysis atmospheric structure through a parameterization- albeit one where aerosol-cloud interactions are not being represented. It is good that the authors explicitly point out that ERA5 clouds are not reanalysis in the same way that things like near surface temperature are.

We agree, thanks for the comment.

Ln 108: Why isn't it also explainable by low CCN? Nd is a function of CCN and updraft and the constant CCN could be unrealistically low.

In the ICON (low CCN) simulation, CCN concentrations are actually not that low for marine regions in absolute terms (250 cm$^{-3}$), but they are low compared to the concentrations in ICON (high CCN), which are set to 1700 cm$^{-3}$. When comparing the CCN concentrations to the cloud droplet number concentrations (Fig 2b), one sees that only 5% of CCN are activated on average, which is very little. By checking the CCN activation parameterization, we conclude that this is due to low vertical velocities at cloud base.
We added these explanations to the methods (Ln 454/ Ln 547): *"For the sake of simplicity, these 2 simulations are referred to as `ICON (low CCN)' and `ICON (high CCN)' in this article, even though `low CCN' can be misleading since 250 cm$^{-3}$ corresponds to typical CCN concentrations in stratocumulus-topped marine boundary layers (Roberts et al., 2010; Allen et al., 2011; Wang et al., 2022; Howes et al., 2023)."* These citations analyze campaign data and also show that activation rates are usually much larger than 5%.

Figure 4: Would be good to note that ERA5 RWC and LWC are the reanalysis thermodynamic fields run through a parameterization.

We have added a sentence concerning the parameterization of LWC and RWC in the discussion of Fig. 4 (Ln 131/Ln 149).

Line 171: Perhaps cite (Karset et al. 2020)

Thanks for the suggestion. This reference fits very well for the description of size-dependent entrainment effects and how that is taken into account (or not) by models, so we added it to the manuscript as mentioned above.

Figure 7 and Table 1: I am not familiar with the assumptions underlying causal graphs and it would help readers such as myself to understand and interpret these results if there was some discussion of how turning off arrows in the model like CCN being able to interact or size-dependent entrainment that we respectively know and suspect exist in the real world affects the causal graph. This seems like a broad discussion of comparing models, which will always be somewhat structurally incomplete to reality.

The causal arrows need to represent actual processes that occur (or are suspected to occur) in the considered data set. For this reason, it does not make sense to have an arrow from $r_{eff}$ to $w_e$ in this model because it does not parameterize size-dependent entrainment enhancement, even though this process does occur in reality, explaining why it is included in the satellite graph. Of course there is some level of subjectivity, as the person drawing the graph is deciding which arrows to include and which ones not to include. However, these choices need to be justified by physical arguments. We have added the following explanation at the beginning of section 3.1 (Ln 218/Ln 247):

"*Fig. 7 shows the assumed $N_d$-LWP causal graph used in the analyses. A causal graph encodes domain knowledge by qualitatively describing causal relationships (arrows) between variables of interest (nodes). Each directed arrow describes a causal effect and not merely a statistical association, and can be justified by an underlying physical process. The colors of the arrows show the magnitude and sign of the direct causal effects $\alpha_{X_i,X_j,I_{ij}}$ calculated from the causal graph.*"

The simplest way to understand what removing an arrow means statistically is by comparing to cloud-controlling factor analyses, such as the ones in Wall et al. (2022): by turning off the arrow pointing from a variable X to a variable Y, we are essentially saying that X does not need to be considered in the multivariate regression of Y with respect to its controlling factors. While multivariate regressions used in the aerosol-cloud literature (e.g. Wall et al., 2022 or Andersen et al., 2016) use varying cloud-controlling factors without necessarily justifying the choices other than saying that these variables are known to 'control' clouds, the causal graph allows to explicitly and very transparently justify why controlling factors are included. We added this short explanation to the methods section (Ln 532/Ln 634).

Concerning subjectivity of the arrow choices: Sensitivity studies can potentially be conducted to evaluate the effect of causal graph assumptions on the results. We did that to some extent in Fons et al. (2023). We would encourage other scientists in the field to come up with their own assumed causal graphs to be compared to ours. We believe this would generate interesting scientific discussions. We briefly discuss the dependence of the results on the assumed causal graph in the conclusion section.

Section 4: This section does a good job summarizing the results of this analysis.

Thank you!

References:
Christensen, M. W., and Coauthors, 2022: Opportunistic experiments to constrain aerosol effective radiative forcing. Atmos. Chem. Phys., 22, 641–674, https://doi.org/10.5194/acp-22-641-2022.

Gryspeerdt, E., and Coauthors, 2019: Constraining the aerosol influence on cloud liquid water path. Atmos. Chem. Phys., 19, 5331–5347, https://doi.org/10.5194/acp-19-5331-2019.

Karset, I. H. H., A. Gettelman, T. Storelvmo, K. Alterskjær, and T. K. Berntsen, 2020: Exploring impacts of size-dependent evaporation and entrainment in a global model. Journal of Geophysical Research: Atmospheres, 125, e2019JD031817.

McCoy, D. T., P. Field, H. Gordon, G. S. Elsaesser, and D. P. Grosvenor, 2020: Untangling causality in midlatitude aerosol–cloud adjustments. Atmos. Chem. Phys., 20, 4085–4103, https://doi.org/10.5194/acp-20-4085-2020.

Stevens, B., and G. Feingold, 2009: Untangling aerosol effects on clouds and precipitation in a buffered system. Nature, 461, 607–613.

Wall, C. J., J. R. Norris, A. Possner, D. T. McCoy, I. L. McCoy, and N. J. Lutsko, 2022: Assessing effective radiative forcing from aerosol–cloud interactions over the global ocean. Proceedings of the National Academy of Sciences, 119, e2210481119, https://doi.org/10.1073/pnas.2210481119.

Wood, R., 2012: Stratocumulus Clouds. Mon. Weather Rev., 140, 2373–2423, https://doi.org/10.1175/MWR-D-11-00121.1.

Wood, R., D. Leon, M. Lebsock, J. Snider, and A. D. Clarke, 2012: Precipitation driving of droplet concentration variability in marine low clouds. J Geophys Res-Atmos, 117, n/a-n/a, https://doi.org/10.1029/2012jd018305.

---

## Author Comment (AC2)

**Authors' answers to Anonymous Referee #2**

We thank the Anonymous Referee #2 for their valuable feedback on our manuscript, which improved the quality of the manuscript significantly. We answer their comments below, using blue font. Where necessary, we indicate the line numbers (Ln) of edits made to the manuscript like so: (Ln in original manuscript/Ln in updated manuscript).

Also please note that we found a tiny mistake in the standardization of the data and re-ran the analyses but the great majority of results remain unchanged: only one arrow in the graph goes from insignificant to significant, but it does not change sign. We added a sentence describing this arrow in the text.

Summary

The authors present a statistical analysis of the drivers of aerosol-cloud interactions in marine boundary layer clouds in satellite observations and in ICON Sappire, a cutting-edge global storm-resolving model.  The statistical analysis aims to get at causal pathways driving changes in cloud properties and dynamics due to aerosol perturbations.  As the aerosol/microphysics/radiation treatment and coupling in the model is somewhat simplified, not all of the same pathways exist in the two frameworks.  However, the analysis yields some insight into the behavior of the model and its biases.

Recommendation: Minor revisions

This paper is advancing a new and seemingly promising statistical technique for understanding the (causal) mechanisms driving aerosol-cloud interactions.  I am supportive of this paper being published in ACP, though after minor revisions.  In the report below, I ask many questions about the causal approach and how the details or uncertainties in the satellite retrievals might affect comparisons with the model results.  As I am not an expert in either statistical analyses related to causal analysis or satellite retrievals, I would ask the authors to think of them as questions driven by curiosity rather than as criticism of the manuscript and study.  Perhaps, other readers new to these approaches might have the same kinds of questions.  Some of them may not be worth addressing in the manuscript.  Most all of the others might just require a reference or a sentence to address.

I would be willing to review the manuscript again if the editor thought it would be useful.

===========================

Major comments:

1. (p. 12/sec 3) I have some knowledge of cloud-aerosol interactions but little of causal approaches such as used in this study.  I have certainly learned a lot about

them by reading this paper and some of its references.  My questions below may be simple minded, but I might not be the only reader to have such questions.  While addressing some of the questions might require a bit of extra exposition, I think only a sentence or two in the appropriate places would be sufficient.

Thanks to the comments from both reviewers, we have realized the description of causal methods was a bit terse. As explained in the Author's comments to Reviewer #1, we have streamlined the discussion of correlative vs. causal satellite studies in the introduction and added a few sentences at the beginning of section 3 to better introduce causal methods. We also added more details to the methods section.

 - My understanding from Fons et al (2023, specifically the caption to supplemental figure 12) is that the satellite causal graph reflects only daytime conditions.  Is that true?  If so, please state it clearly somewhere.  (Apologies if I've missed that.)  However, the availability of both day- and night-time data from ICON offers the possibility to see whether the causal networks change from day to night (at least in model world).  That seems interesting, because marine boundary layer clouds have significant diurnal cycles of radiative cooling, entrainment, liquid water path and precipitation, which could (?) change the causal connections between day and night conditions.

That is a very good point, we forgot to add this discussion in the manuscript, thank you for the suggestion. The satellite retrievals for optical cloud properties only include daytime data points as they are based on radiance measurements in visible/near-IR channels. We have added a sentence mentioning this in the methods section (Ln 497/Ln 595).

The model results presented in the pre-print included both daytime and nighttime data. We additionally provide comparative daytime vs. nighttime results for the model in the updated version of the manuscript (Fig. 8, Ln 369/Ln 413-428). These comparative results nicely illustrate that adjustments will be different at night and during the day, implying a selection bias for daytime-only satellite studies.

We also now explicitly indicate in the figure captions whether the plotted reanalysis/model data correspond to daytime only or both daytime and nighttime, depending on the variable.

 - As GPM satellites with radar and/or microwave instruments are not continuously overhead, most precipitation data will rely on the same geostationary satellite that is also providing information about cloud optical depth, effective radius and cloud top temperature.  When combined with the adiabatic assumption, there seem to be many fewer degrees of freedom in the satellite data than arrows in the causal diagram.  How should the reader think about this?

We discussed this issue briefly in Fons et al. (2023). $N_d$, $r_{eff}$, $H$ and LWP come from the same radiance measurements in the visible and near-infrared. Precipitation will be inferred from infrared measurements from the same satellite instrument, combined with polar-orbiting microwave measurements propagated along wind vectors. Arola et al. (2022) (https://doi.org/10.1038/s41467-022-34948-5) found that correlated retrieval errors in $r_{eff}$/COD satellite products, associated with the adiabatic assumptions, could result in spurious $N_d$-LWP associations. As you mentioned, this could also be true to some extent for precipitation measurements that come from the same satellite, despite coming from different wavelength channels. We have now added a sentence describing the impact of correlated noises in the conclusion (Ln 415/Ln 502).

 - Rainfall (measured at cloud base or the surface) matters to the dynamics of the boundary layer when it is strong and has little influence when it is weak (in comparison to the other drivers of turbulence and convection).  If the impact of rainfall on the boundary layer is nonlinear, how will this show up in the causal relationships, which seem to be linear and based on the standardized changes of rainfall across all times?

It is true that the linear assumption could explain why the behavior of RR in the causal graph is not completely as expected. We have mentioned this in the discussion of the direct causal effects. If you could suggest a reference for the non-linearities you describe, we would be happy to add it to the discussion. For the moment, we have added a description of the precipitation arrows that will be impacted by the linear assumption (Ln 256/Ln 327): "*It should also be noted that the causal method used here is based on a linearity assumption which might be broken for non-linear precipitation processes. In particular, the lagged arrow from RR to $N_d$ (wet scavenging and dynamical effects of precipitation) might be incorrectly captured in case of non-linear threshold effects of precipitation on the dynamics of the boundary layer. The onset of precipitation (arrow from $r_{eff}$ to RR) is also non-linear but is expected to follow a power law, thus it should be well captured thanks to the log-transformation of the variables (see methods)*".

 - As susceptibility of low cloud quantities is often defined in terms of logarithmic changes (e.g., equation 4 in Terai et al, 2012, https://doi.org/10.5194/acp-12-4567-2012), would it be worth contrasting that briefly with the linear sensitivities represented by the alpha's and beta's in this study (although standardized as described as in Fons et al, 2003)?

As in Fons et al. (2023), the cloud variables are log-transformed in line with other studies in the literature, but we did forget to mention the log-transformation in the methods, thanks for catching the omission. This has now been added (Ln 520/Ln 619).

- Cloud fraction adjustments to increased aerosol occur in clouds with low background aerosol concentrations.  Are they worth including in the causal network, especially if the pixel size for the causal analysis is 0.5 degrees?

The 0.5 ° average is performed on cloudy pixels only, so CF adjustments should not be captured by the data. We have also added this to the methods section (Ln 520/ Ln 617).

 - Are the causal relationships in the network based on the high CCN or the low CCN simulation, or a combination of the two?  My understanding of the time series analysis is that the causal relationships depend on time variations in predictor quantities in a single simulation.  That seems quite different from other approaches that difference quantities between the high and low CCN simulations, so it would be good to be clear on this point.

That is a good distinction to make, we have added this to the methods section. The causal analyses are based on the 'low CCN' simulation, we have also added this to the methods section (Ln 538/Ln 645).

 - (inspired by p. 17/fig 8) If the causal effects are computed from time series with an increment in predictor (e.g., aerosols) at time zero, wouldn't the influence of that local in time-and-space aerosol increase move away from a fixed Eulerian location as the winds advect the airmass?  What is the meaning of these after-effects 24 hours later than an aerosol (or other) perturbation at a fixed Eulerian location?  Does the aggregation of data to larger scales (here 0.5 degrees and 10 degrees in Fons et al, 2023) avoid this issue?

We discussed this in Fons et al. (2023) but added a brief discussion in this paper as well (Ln 331/Ln 373). The causal temporal developments are not calculated from an observed temporal evolution. Instead, they are calculated from the time propagation of the direct causal effects and autodependency coefficients (shown in Fig. 7). The direct causal effects are computed over 15-min intervals only, and we assume that there is minimal advection of cloud fields past the 0.5° boxes within this time frame. The resulting propagated temporal developments should be understood as hypothetical temporal developments should the clouds persist for this long, but they are not a direct measurement of cloud lifetimes.

 - Also, do satellite retrievals of effective radius correspond to cloud top values or some type of integral over the depth of the cloud?  Shang et al (2023, ACP, https://doi.org/10.5194/acp-23-2729-2023) suggest that different wavelengths may give information about different vertical levels.

$r_{eff}$ satellite retrievals correspond to the effective radius close to cloud top. To be exact, this is only an approximation, but King et al., (1992) estimate that $r_{eff}$ should correspond to the droplet radius at 85-95% of the cloud height for optically thick

clouds, i.e., close to cloud top. We added 'cloud effective radius *at cloud top'* to the methods to make this clear (Ln 496/Ln 593).

If answers to these questions are in references like Fons et al (2023), Runge et al (2019) or elsewhere, the authors could emphasize that in the text.

============================

Specific/minor comments (11/240 means p. 11, line 240):

 - Perhaps, the abstract could mention that the simulations are made for a period during boreal summertime.

This has now been added.

5/109: Did ICON try to include an estimate for subgrid vertical velocity variance in the parameterization for activation?  While subject to uncertainty, that might have helped correct the low bias in Nd.  However, this might not be worth such an effort when using a Smagorinsky-type parameterization, because they do not perform well in predicting subgrid quantities in simulations with these kinds of grid spacings.  See e.g., Cheng et al (2010, https://doi.org/10.3894/JAMES.2010.2.3, sec. 3.1.2.1).

The activation parameterization used here does not include an estimate of sub-grid scale velocity variance. Propositions for how to include such an estimate to ICON-HAM km-scale set-ups have been discussed at the latest HAMMOZ workshop, but this is being the scope of our study.

8/Fig 4: Do metrics related to decoupling change between the low and high CCN simulations?  The drizzling boundary layer might encourage such decoupling with rain evaporation below cloud base.  The differing rain mixing ratios (and presumably) rain rates suggest different levels of precipitation-related subcloud evaporation in the two simulations.

We have evaluated boundary layer decoupling by looking at the height difference between the cloud base (CB) and the LCL, as suggested by Jones et al (2011) (doi:10.5194/acp-11-7143-2011). We find that most boundary layers are not decoupled, and that the high CCN simulation shows slightly increased decoupling compared to the low CCN simulation despite its lower rain rates. This might be due to the deeper boundary layer and higher entrainment rates in the high CCN simulation. However, we also note that, due to the relatively coarse resolution of the model around the inversion (100 + meters), it might be difficult to accurately estimate boundary layer decoupling as the height difference between the LCL and the cloud-base. Similarly, due to the coarse resolution, the inversion is not sharp enough, making it difficult to use the other decoupling metrics suggested by Jones et al. (2011). We have nonetheless added a supplementary Figure showing the CB-

LCL decoupling metric for the low vs. high CCN simulation for all four regions and we refer to it in the discussion of Fig. 4 (Ln 125/Ln 137-140).

In the outlook, we also added decoupling as an example of non-linear processes that could be mis-estimated with the linear causal approach (Ln 416 / Ln 504): *"Non-linear causal methods could be a good option to better estimate precipitation effects but also understand how potentially non-linear decoupling of the boundary layer can modulate the results presented here."*

Also, vertical velocity variance might be useful for thinking about activation and cloud formation, alongside the mean vertical velocity.

Fig. 2g gives an idea of the variance in cloud-base vertical velocity in the 0.5° x 0.5° averaged data. The variance in the original 5-km data is about 3 times the one in the coarser data. We have added a supplementary figure showing the same as Fig. 2g but using the original 5-km resolution data. Even in that figure, we see that the 75th percentile of cloud-base velocity is on the order of magnitude of a few 10s of mm.s$^{-1}$, which is quite low and confirms the argument that low updraft speeds are responsible for the low cloud droplet number concentrations.

9/Fig 5: The labels a/b in the caption refer to the wrong panels.  Also, why not also show the same plot for the low CCN simulation, or at least a contour showing how the joint Nd-LWP distribution shifts from low to high CCN?

Thank you for catching the labeling mistake, we fixed it. We also added a cross-reference to Suppl. Fig. 2 where histograms of the low CCN simulation are provided as well. The reasoning for only showing the high CCN 2D histograms in the main text is that cloud droplet number concentrations are very low in the low CCN simulation, preventing a fair comparison to the satellite data histogram.

Is the joint distribution based on grid point/satellite retrieval footprint data?  if one or both datasets coarsened to the 0.25 or 0.5 degree resolutions mentioned in the satellite data appendix, a short description of how the Nd or effective radius data are coarsened (e.g., average over all cloudy columns) would be useful.

We added the data resolution (0.5°) to the caption, and the methods now include a mention of the averaging method, which is in-cloud, not all-sky (Ln 517/Ln 617).

With the high CCN simulation having uniform "aerosol" concentrations, I would expect that changes in cloud base Nd in ICON are induced mainly by changes in the updraft strength, while satellite Nd changes are most likely dominated by actual aerosol variability.  Is this consistent with the interpretation of the authors?  If so, how should this contrast make us think about the results?

As alluded to on Ln 150/Ln169, the idea for this study is that we use the standard partial sensitivity decomposition for the LWP adjustment (e.g. Bellouin 2020). With these causal graphs, we focus on the $\partial$LWP/$\partial$N$_d$ term, but ignore the $\partial$N$_d$/$\partial$Aerosol term, which this comment is about. For this term, it does not matter why $N_d$ changes, as long as it changes. We added this after discussing the term separation on Ln150/Ln 169.

"Separating the terms and focusing only on the first one also allows for a fairer comparison of the satellite and model data, as the model's $\partial$N$_d$/$\partial$A term will not be defined due to the constant CCN assumption"

9/163: The "invisible ship tracks" paper of Manshausen et al (2022, https://doi.org/10.1038/s41586-022-05122-0) does show positive LWP adjustments in trade cumulus regions and might be worth mentioning here alongside Jiang et al.

Great suggestion. We have added the citation to Manshausen et al. (2022), but rather in the conclusion of the manuscript (Ln 405/Ln 485):

"In fact, Manshausen et al. (2022) used ship tracks to show that LWP adjustments are weakly negative in the stratocumulus cloud regimes, but positive in the trade cumulus cloud regime where most tracks are `invisible'. This is consistent with the cumuliform-looking ICON clouds showing positive LWP adjustments in our study.'"

10/Fig 6: Are the Nd and r_d values shown here averaged over all grid points (cloudy and clear) or only over cloudy grid points? The in-cloud values are a more useful reference when thinking about aerosol-cloud interactions. I would be surprised if the in-cloud values of N_d and r_d would go to zero so smoothly at the top and bottom of the cloud layer, but perhaps this does happen in ICON.

Fig. 6 shows all sky values, as they look nicer for plotting vertical profiles. We had explicitly mentioned this for the vertical profiles in Fig. 4 (see Ln 160-165). We had also included Suppl. Fig. 3, showing instantaneous cloudy-pixel-only values as histograms. We have now added instantaneous $N_d$ and $r_d$ values in Suppl. Fig. 3 and we now mention the all-sky average and Suppl. Fig. 3 when discussing Fig. 6 as well.

11/182 and 14/266: Does ICON Sapphire really use a fixed effective radius for all cloud liquid even when the double moment microphysics is used? I would understand this choice, but this should really be explicitly stated somewhere in this paper since I couldn't find it in the original Sapphire paper.

The radius can be diagnosed from the prognostic cloud liquid water content and the cloud droplet number concentration given assumptions about the size distribution. However, we never wanted to imply that $r_d$ is constant. What we wanted to say here is that cloud droplet sedimentation is not parameterized because, due to the vertical extent of the grid boxes, cloud droplet sedimentation is expected to be a

negligible sink for cloud water compared to evaporation or rain autoconversion. We added this explanation of why cloud droplet sedimentation is turned off in the manuscript (Ln 181/Ln 207).

13/236: Could the magnitudes in the model graph be stronger because the model Nd is low and the clouds in the model simulation are precipitating more strongly than in the satellite data?  Is it clear that the alpha's should be invariant as the background aerosol/Nd changes?

If the relationships are indeed linear, then the alphas should be invariant. However, if they are not in reality linear, the alphas could vary. We had discussed this linearity assumption in the analyses and in the discussion but not explicitly at this specific location of the manuscript. We added the following sentence after mentioning the different magnitudes in the model and in the satellite data: "Another explanation could be that the ICON graph captures a non-linear lower-$N_d$/higher-RR regime, potentially causing the alphas to vary in magnitude due to the linear assumption behind Wright's path analysis." (Ln 237/Ln 270)

top of p. 14: For the reader, it might be helpful to first talk about alpha's with clear signals in the satellite and model before later talking about those that are more complicated.

We moved the discussion of precipitation influences, which are less well quantified by the causal approach, to after the discussion of entrainment influences, which are more straightforward.

15/304: Stevens and Seifert (2008, https://doi.org/10.2151/jmsj.86A.143) is an earlier study along these lines.  Albrecht (1993, https://doi.org/10.1029/93JD00027) isn't a direct predecessor but makes clear the impact of precipitation on the depth of the marine boundary layer.

Thank you for the suggestion, we have added the citation to Stevens and Seifert (2008). (Ln 304/Ln 344)

19/420: Possible reference for "increasing the vertical resolution": Bogenschutz et al (2023, https://doi.org/10.5194/gmd-16-335-2023).  More sophisticated subgrid turbulence closures also offer some promise of better representation of stratocumulus at kilometer-scale grid spacing: Shi et al (2018, https://doi.org/10.1175/JAS-D-17-0162.1), Bogenschutz et al (2023, https://doi.org/10.1029/2022MS003466).

Thank you for the suggestions! We have added these references.

20/449: Specify a particular altitude, not one relative to the inversion height (which is variable).

We modified the sentence to be more descriptive of the actual vertical resolution (Ln 449/ Ln 540): *"The vertical resolution increases progressively from 20 m at the surface to 400 m at an altitude of 8 km, with vertical resolutions around 125 - 160 m at the inversion level."*

22: It would be worth mentioning the footprint of the satellite retrievals from SEVIRI, to give some context about how many pixels are being averaged into the 0.25 or 0.5 degree grids.

This has been added on Ln493/Ln588.

===========================

Typographical/rephrasing suggestions (OPTIONAL):

12/215: "along" --> "alongside"

16/318: "datam" --> "data"

19/421: "e.g.," should be before the reference.  Maybe add another <> before the {PossnerEtAl2014}?

20/444: "a an" --> "an"
**Citation**: https://doi.org/10.5194/egusphere-2024-195-RC2

These typos have been corrected, thank you for catching them.